# FEDERATED SEMI-SUPERVISED LEARNING WITH INTER-CLIENT CONSISTENCY & DISJOINT LEARNING

**Wonyong Jeong[1], Jaehong Yoon[2], Eunho Yang[1,3], and Sung Ju Hwang[1,3]**
Graduate School of AI[1], KAIST, Seoul, South Korea
School of Computing[2], KAIST, Daejeon, South Korea
AITRICS [3], Seoul, South Korea
{wyjeong, jaehong.yoon, eunhoy, sjhwang82}@kaist.ac.kr

## ABSTRACT

While existing federated learning approaches mostly require that clients have fully-labeled data to train on, in realistic settings, data obtained at the client-side often comes without any accompanying labels. Such deficiency of labels may result from either high labeling cost, or difficulty of annotation due to the requirement of expert knowledge. Thus the private data at each client may be either partly labeled, or completely unlabeled with labeled data being available only at the server, which leads us to a new practical federated learning problem, namely *Federated Semi-Supervised Learning* (FSSL). In this work, we study two essential scenarios of FSSL based on the location of the labeled data. The first scenario considers a conventional case where clients have both labeled and unlabeled data (labels-at-client), and the second scenario considers a more challenging case, where the labeled data is only available at the server (labels-at-server). We then propose a novel method to tackle the problems, which we refer to as *Federated Matching* (FedMatch). FedMatch improves upon naive combinations of federated learning and semi-supervised learning approaches with a new inter-client consistency loss and decomposition of the parameters for disjoint learning on labeled and unlabeled data. Through extensive experimental validation of our method in the two different scenarios, we show that our method outperforms both local semi-supervised learning and baselines which naively combine federated learning with semi-supervised learning. The code is available at https://github.com/wyjeong/FedMatch.

## 1 INTRODUCTION

*Federated Learning (FL)* (McMahan et al., 2017; Zhao et al., 2018; Li et al., 2018; Chen et al., 2019a;b), in which multiple clients collaboratively learn a global model via coordinated communication, has been an active topic of research over the past few years. The most distinctive difference of federated learning from distributed learning is that the data is only *privately accessible* at each local client, without inter-client data sharing. Such decentralized learning brings us numerous advantages in addressing real-world issues such as data privacy, security, and access rights. For example, for on-device learning of mobile devices, the service provider may not directly access local data since they may contain privacy-sensitive information. In healthcare domains, the hospitals may want to improve their clinical diagnosis systems without sharing the patient records.

Existing federated learning approaches (McMahan et al., 2017; Wang et al., 2020; Li et al., 2018) handle these problems by aggregating the locally learned model parameters. A common limitation is that they only consider supervised learning settings, where the local private data is fully labeled. Yet, the assumption that all of the data examples may include sophisticate annotations is not realistic for real-world applications. Suppose that we perform on-device federated learning, the users may not want to spend their time and efforts in annotating the data, and the participation rate across the users may largely differ. Even in the case of enthusiastic users may not be able to fully label all the data in the device, which will leave the majority of the data as unlabeled **(See Figure 1 (a))**. Moreover, in some scenarios, the users may not have sufficient expertise to correctly label the data. For instance, suppose that we have a workout app that automatically evaluates and corrects one's body posture. In this case, the end users may not be able to evaluate his/her own body posture at all **(See Figure 1**

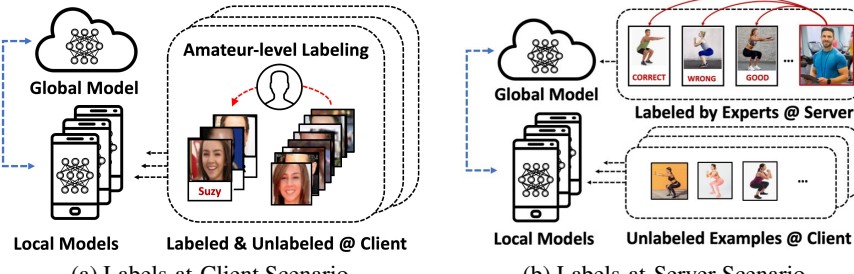

Figure 1: **Illustrations of Two Practical Scenarios in Federated Semi-Supervised Learning** (a) Labels-at-Client scenario: both labeled and unlabeled data are available at local clients. (b) Labels-at-Server scenario: labeled instances are available only at server, while unlabeled data are available at local clients.

**(b))**. Thus, in many realistic scenarios for federated learning, local data will be mostly *unlabeled*. This leads us to practical problems of federated learning with deficiency of labels, namely *Federated Semi-Supervised Learning* (FSSL).

A naive solution to these scenarios is to simply perform Semi-Supervised Learning (SSL) using any off-the-shelf methods (e.g. FixMatch (Sohn et al., 2020), UDA (Xie et al., 2019)), while using federated learning algorithms to aggregate the learned weights. Yet, this does not fully exploit the knowledge of the multiple models trained on heterogeneous data distributions. To address this problem, we present a novel framework which we refer to as *Federated Matching (FedMatch)*, which enforces the consistency between the predictions made across multiple models. Further, conventional semi-supervised learning approaches are not applicable for scenarios where labeled data is only available at the server (Figure 1 (b)), which is a unique SSL setting for federated learning. Also, even when the labeled data is available at the client (Figure 1 (a)), learning from the unlabeled data may lead to forgetting of what the model learned from the labeled data. To tackle these issues, we *decompose* the model parameters into two, a dense parameter for supervised and a sparse parameter for unsupervised learning. This sparse additive parameter decomposition ensures that training on labeled and unlabeled data are effectively separable, thus minimizing interference between the two tasks. We further reduce the communication costs with both the decomposed parameters by sending only the difference of the parameters across the communication rounds. We validate FedMatch on both scenarios (Figure 1 (a) and (b)) and show that our models significantly outperform baselines, including a naive combination of federated learning with semi-supervised learning, on the training data which are both non-i.i.d. and i.i.d. data. The main contributions of this work are as follows:

- We introduce a practical problem of federated learning with deficiency of supervision, namely **Federated Semi-Supervised Learning (FSSL)**, and study two different scenarios, where the local data is partly labeled (*Labels-at-Client*) or completely unlabeled (*Labels-at-Server*).

- We propose a **novel method**, **Federated Matching (FedMatch)**, which learns inter-client consistency between multiple clients, and decomposes model parameters to *reduce* both interference between supervised and unsupervised tasks, and communication cost.

- We show that our method, FedMatch, **significantly outperforms** both local SSL and the naive combination of FL with SSL algorithms under the conventional labels-at-client and the **novel labels-at-server** scenario, across multiple clients with both non-i.i.d. and i.i.d. data.

## 2 PROBLEM DEFINITION

We begin with formal definition of Federated Learning (FL) and Semi-Supervised Learning (SSL). Then, we define Federated Semi-Supervised Learning (FSSL) and introduce two essential scenarios.

### 2.1 PRELIMINARIES

**Federated Learning**   Federated Learning (FL) aims to collaboratively learn a global model via coordinated communication with multiple clients. Let $G$ be a global model and $\mathcal{L} = \{l_k\}_{k=1}^{K}$ be a set of local models for $K$ clients. Let $\mathcal{D} = \{\mathbf{x}_i, \mathbf{y}_i\}_{i=1}^{N}$ be a given dataset, where $\mathbf{x}_i$ is an arbitrary training instance with a corresponding one-hot label $\mathbf{y}_i \in \{1, \ldots, C\}$ for the $C$-way multi-class classification problem and $N$ is the number of instances. $\mathcal{D}$ is composed of $K$ sub-datasets $\mathcal{D}^{l_k} = \{\mathbf{x}_i^{l_k}, \mathbf{y}_i^{l_k}\}_{i=1}^{N^{l_k}}$ privately collected at each client or local model $l_k$. At each communication round $r$, $G$ first randomly selects $A$ local models that are available for training $\mathcal{L}^r \subset \mathcal{L}$ and $|\mathcal{L}^r| = A$. The global model $G$ then initializes $\mathcal{L}^r$ with global weights $\boldsymbol{\theta}^G$, and the active local models $l_a \in \mathcal{L}^r$ perform supervised

learning to minimize loss $\ell_s(\boldsymbol{\theta}^{l_a})$ on the corresponding sub-dataset $\mathcal{D}^{l_a}$. $G$ then aggregates the learned weights $\boldsymbol{\theta}^G \leftarrow \frac{N^{l_a}}{N} \sum_a^A \boldsymbol{\theta}^{l_a}$ and broadcasts newly aggregated weights to local models that would be available at the next round $r + 1$, and repeat the learning procedure until the final round $R$.

**Semi-Supervised Learning**  Semi-supervised learning (SSL) refers to the problem of learning with partially labeled data, where the ratio of unlabeled data is usually much larger than that of the labeled data (e.g. $1 : 10$). For SSL, $\mathcal{D}$ is further split into labeled and unlabeled data. Let $\mathcal{S} = \{\mathbf{x}_i, \mathbf{y}_i\}_{i=1}^S$ be a set of $S$ labeled data instances and $\mathcal{U} = \{\mathbf{u}_i\}_{i=1}^U$ be a set of $U$ unlabeled samples without corresponding label. Here, in general, $|\mathcal{S}| \ll |\mathcal{U}|$. With these two datasets, $\mathcal{S}$ and $\mathcal{U}$, we now perform semi-supervised learning. Let $p_{\boldsymbol{\theta}}(\mathbf{y}|\mathbf{x})$ be a neural network that is parameterized by weights $\boldsymbol{\theta}$ and predicts softmax outputs $\hat{\mathbf{y}}$ with given input $\mathbf{x}$. Our objective is to minimize loss function $\ell_{final}(\boldsymbol{\theta}) = \ell_s(\boldsymbol{\theta}) + \ell_u(\boldsymbol{\theta})$, where $\ell_s(\boldsymbol{\theta})$ is loss term for supervised learning on $\mathcal{S}$ and $\ell_u(\boldsymbol{\theta})$ is loss term for unsupervised learning on $\mathcal{U}$

## 2.2 Federated Semi-Supervised Learning

Now we further describe a practical problem of deficiency of labels in federated learning, which we refer to as Federated Semi-Supervised Learning (FSSL), in which the data obtained at the clients may or may not come with accompanying labels. Given a dataset $\mathcal{D} = \{\mathbf{x}_i, \mathbf{y}_i\}_{i=1}^N$, $\mathcal{D}$ is split into a labeled set $\mathcal{S} = \{\mathbf{x}_i, \mathbf{y}_i\}_{i=1}^S$ and a unlabeled set $\mathcal{U} = \{\mathbf{u}_i\}_{i=1}^U$ as in the standard semi-supervised learning. Under the federated learning framework, we have a global model $G$ and a set of local models $\mathcal{L}$ where the unlabeled dataset $\mathcal{U}$ is privately spread over $K$ clients hence $\mathcal{U}^{l_k} = \{\mathbf{u}_i^{l_k}\}_{i=1}^{U^{l_k}}$. For a labeled set $\mathcal{S}$, on the other hand, we consider two different scenarios depending on the availability of labeled data at clients, namely the *Labels-at-Client* and the *Labels-at-Server* scenario, of which problem settings and learning procedures will be discussed later.

## 3 Federated Matching

We now describe our *Federated Matching (FedMatch)* algorithm to tackle the federated semi-supervised learning problem. We describe its core components in detail in the following subsections.

## 3.1 Inter-Client Consistency Loss

Consistency regularization (Xie et al., 2019; Sohn et al., 2020; Berthelot et al., 2019b;a) is one of most popular approaches to learn from unlabeled examples in a semi-supervised learning setting. Conventional consistency-regularization methods enforce the predictions from the augmented examples and original (or weakly augmented) instances to output the same class label, $||p_{\boldsymbol{\theta}}(\mathbf{y}|\mathbf{u}) - p_{\boldsymbol{\theta}}(\mathbf{y}|\pi(\mathbf{u}))||_2^2$, where $\pi(\cdot)$ is a stochastic transformation function. Based on the assumption that class semantics are unaffected by small input perturbations, these methods basically ensures consistency of the prediction across the multiple perturbations of *same input*. For our federated semi-supervised learning method, we additionally propose a novel consistency loss that regularizes the *models* learned at multiple clients to output the same prediction. This novel consistency loss for FSSL, *inter-client consistency* loss, is defined as follows:

$$\frac{1}{H} \sum_{j=1}^H \text{KL}[p_{\boldsymbol{\theta}^{h_j}}^*(\mathbf{y}|\mathbf{u})||p_{\boldsymbol{\theta}^l}(\mathbf{y}|\mathbf{u})] \tag{1}$$

where $p_{\boldsymbol{\theta}^h}^*(\mathbf{y}|\mathbf{x})$ are helper agents that are selected from the server based on model similarity to the client (which we describe later), that are not trained at the client (* denotes that we freeze the parameters). The server selects and broadcasts $H$ helper agents at each communication round. We also use data-level consistency regularization at each local client similarly to FixMatch (Sohn et al., 2020). Our final consistency regularization term $\Phi(\cdot)$ can be written as follows:

$$\Phi(\cdot) = \text{CrossEntropy}(\hat{\mathbf{y}}, p_{\boldsymbol{\theta}^l}(\mathbf{y}|\pi(\mathbf{u}))) + \frac{1}{H} \sum_{j=1}^H \text{KL}[p_{\boldsymbol{\theta}^{h_j}}^*(\mathbf{y}|\mathbf{u})||p_{\boldsymbol{\theta}^l}(\mathbf{y}|\mathbf{u})] \tag{2}$$

where $\pi(\mathbf{u})$ performs RandAugment (Cubuk et al., 2019) on unlabeld instance $\mathbf{u}$. $\hat{\mathbf{y}}$ is our novel pseudo-labeling technique, which we refer to as the *agreement-based* pseudo label, defined as follows:

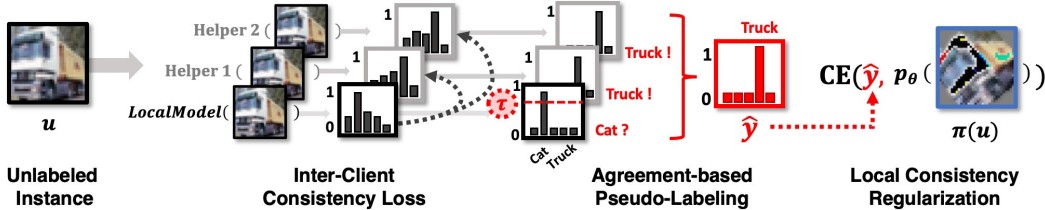

Figure 2: **Illustration of Inter-Client Consistency Loss.** We illustrate each step of our inter-client consistency regularization process performed at local client. We provide the detailed explanations in Section 3.1.

$$\hat{\mathbf{y}} = \text{Max}(\mathbb{1}(p^*_{\boldsymbol{\theta}^l}(\mathbf{y}|\mathbf{u})) + \sum_{j=1}^{H} \mathbb{1}(p^*_{\boldsymbol{\theta}^{h_j}}(\mathbf{y}|\mathbf{u}))) \tag{3}$$

where $\mathbb{1}(\cdot)$ produces one-hot labels with given softmax values , and $\text{Max}(\cdot)$ outputs one-hot labels on the class that has the maximum agreements. We discard instances with low-confident predictions below confidence threshold $\tau$ when generating pseudo-labels. We then perform standard cross-entropy minimization with the pseudo-label $\hat{\mathbf{y}}$.

For helper agents, we select the $H$ helper agents $p^*_{\boldsymbol{\theta}^{h_{j:H}}}(\mathbf{y}|\mathbf{u})$ for each client as the most relevant models from other clients. Specifically, we represent each model by its prediction $\mathbf{m}$ on the same arbitrary input $\mathbf{a}$ located at server (we use random Gaussian noise), such that $\mathbf{m}^l = p_{\boldsymbol{\theta}^l}(\mathbf{m}|\mathbf{a})$. The server tries to keep and update all model embeddings $\mathbf{m}^{1:K}$ from clients once each client updates its weights to server, and creates *K-Dimensional Tree (KD Tree)* on $\mathbf{m}$ in the current round $r$ for nearest neighbor search to rapidly select the $H$ helper agents for each client in the next rounds. We send helper agents for every 10 rounds, and if a certain client has not yet updated its weights to server in the previous step, then server simply skips sending helpers to the client at the round.

## 3.2 PARAMETER DECOMPOSITION FOR DISJOINT LEARNING

In the standard semi-supervised learning approaches, learning on labeled and unlabeled data is simultaneously done with a shared set of parameters. However, since this is inapplicable to the disjoint learning scenario (Figure 1 (b)) and may result in forgetting of knowledge of labeled data (see Figure 6 (c)), we separate the supervised and unsupervised learning via the decomposition of model parameters into two sets of parameters, one for supervised learning and another for unsupervised learning. To this end, we decompose our model parameters $\theta$ into two variables, $\sigma$ for supervised learning and $\psi$ for unsupervised learning, such that $\theta = \sigma + \psi$. We perform standard supervised learning on $\sigma$, while keeping $\psi$ fixed during training, by minimizing the loss term as follows:

$$\text{minimize } \mathcal{L}_s(\sigma) = \lambda_s \text{CrossEntropy}(\mathbf{y}, p_{\sigma+\psi^*}(\mathbf{y}|\mathbf{x})) \tag{4}$$

where $\mathbf{x}$ and $\mathbf{y}$ are from labeled set $\mathcal{S}$. For learning on unlabeled data, we perform unsupervised learning conversely on $\psi$, while keeping $\sigma$ fixed for the learning phase, by minimizing the consistency loss terms as follows:

$$\text{minimize } \mathcal{L}_u(\psi) = \lambda_{\text{ICCS}}\Phi_{\sigma^*+\psi}(\cdot) + \lambda_{L_2}||\sigma^* - \psi||_2^2 + \lambda_{L_1}||\psi||_1 \tag{5}$$

where all $\lambda$s are hyper-parameters to control the learning ratio between the terms. We additionally add $L_2$- and $L_1$-Regularization on $\psi$ such that $\psi$ is sparse, while not drifting far from knowledge that $\sigma$ has learned. To sum up, our decomposition technique allows us:

- **Preservation Reliable Knowledge from Labeled Data**: We empirically find that learning on both labeled and unlabeled data with a single set of parameters may result in the model to forget about what it learned from the labeled data (see Figure 6 (c)). Our method can effectively prevent the *inter-task interference* via utilizing disjoint parameters only for supervised learning.

- **Reduction of Communication Costs**: Sparsity on the unsupervised parameter $\psi$ allows to reduce communication cost. In addition, we further minimize the cost by subtracting the learned knowledge for each parameter, such that $\Delta\psi = \psi_r^l - \psi_r^G$ and $\Delta\sigma = \sigma_r^l - \sigma_r^G$, and transmit only the differences $\Delta\psi$ and $\Delta\sigma$ as sparse matrices for both client-to-server and server-to-client costs.

- **Disjoint Learning**: In federated semi-supervised learning, labeled data can be located at either client or server, which requires the model's learning procedure to be flexible. Our decomposition technique allows the model for the supervised training to be done separately elsewhere.

**Algorithm 1 Labels-at-Client Scenario**

1: **RunServer()**
2: initialize $\sigma^0$ and $\psi^0$
3: **for** each round $r = 1, 2, ..., R$ **do**
4:    $\mathcal{L}^r \leftarrow$ (select random $A$ clients from $\mathcal{L}$)
5:    **for** each client $l_a^r \in \mathcal{L}^r$ **in parallel do**
6:       $\psi_{1:H}^r \leftarrow$ GetNearestNeighbors($\psi^r$)
7:       $\sigma_a^r, \psi_a^r \leftarrow$ RunClient($\sigma^r, \psi^r, \psi_{1:H}^r$)
8:       EmbedLocalModel($\sigma_a^r, \psi_a^r$)
9:    **end for**
10:   $\sigma^{r+1} \leftarrow \frac{1}{A}\sum_{a=1}^A (\sigma_{l_a}^r)$
11:   $\psi^{r+1} \leftarrow \frac{1}{A}\sum_{a=1}^A (\psi_{l_a}^r)$
12: **end for**
13: **RunClient($\sigma, \psi, \psi_{1:H}$)**
14: $\theta_{l_a} \leftarrow \sigma + \psi, \theta_{h_{1:H}} \leftarrow \sigma + \psi_{1:H}$
15: **for** each local epoch $e$ from 1 to $E_L$ **do**
16:   **for** minibatch $s \in \mathcal{S}_{l_a}$ and $u \in \mathcal{U}_{l_a}$ **do**
17:     $\theta_{\sigma+\psi^*} \leftarrow \theta_{\sigma+\psi^*} - \eta\nabla\ell_s(\theta_{\sigma+\psi^*}; \theta_{h_{1:H}}, s)$
18:     $\theta_{\sigma^*+\psi} \leftarrow \theta_{\sigma^*+\psi} - \eta\nabla\ell_u(\theta_{\sigma^*+\psi}; \theta_{h_{1:H}}, u)$
19:   **end for**
20: **end for**

Figure 3: **Illustrative Running Example of Labels-at-Client Scenario** We describe training and communication procedure between local and global model under Labels-at-Client scenario corresponding to the Algorithm 1. More details are described in Section 4.

## 4 LABELS-AT-CLIENT SCENARIO

**Problem Definition** The *Labels-at-Client* scenario posits that the end-users intermittently annotate a small portion of their local data (i.e., $5\%$ of the entire data), leaving the rest of data instances unlabeled as illustrated in Figure 1 (a). This is a common scenario for user-generated personal data, where the end-users can easily annotate the data but may not have time or motivation to label all the data (e.g. annotating faces in pictures for photo albums or social networking). We assume that clients train on both labeled and unlabeled data, while the server only aggregates the updates from the clients and redistributes the aggregated parameters back to the clients. In this scenario, labeled data $\mathcal{S}$ is a set of individual sub-datasets $\mathcal{S}^{l_k} = \{\mathbf{x}_i^{l_k}, \mathbf{y}_i^{l_k}\}_{i=1}^{S^{l_k}}$, yielding $K$ sub-datasets for $K$ local models $l_{1:K}$. The overall learning procedure of the global model is the same as that of conventional federated learning (global model $G$ aggregates updates from the selected subset of clients and broadcasts them), except that active local models $l_{1:A}$ perform semi-supervised learning by minimizing the loss $\ell_{final}(\boldsymbol{\theta}^{l_a}) = \ell_s(\boldsymbol{\theta}^{l_a}) + \ell_u(\boldsymbol{\theta}^{l_a})$ respectively on $\mathcal{S}^{l_a}$ and $\mathcal{U}^{l_a}$.

**FedMatch Algorithm for Labels-at-Client Scenario** Now we introduce our *FedMatch* algorithm for the labels-at-client scenario. As shown in Figure 3, which illustrates an example case of the labels-at-client scenario, active local models $l_{1:A}$ at the current round $r$ learn both $\sigma^{l_{1:A}^r}$ and $\psi^{l_{1:A}^r}$ on both the labeled data $\mathcal{S}^{l_{1:A}}$ and unlabeled data $\mathcal{U}^{l_{1:A}}$ at each local environment. After the completion of local training, the clients update both their learned knowledge $\sigma^{l_{1:A}}$ and $\psi^{l_{1:A}}$ to the server. The server then aggregates $\sigma^{l_{1:A}}$ and $\psi^{l_{1:A}}$, *respectively*, after embedding local models based on model similarity as well as create KD-Tree to rapidly retrieve the top-$H$ nearest neighbors $\psi^{h_{1:H}}$ for each client. At the next round, the server transmits the aggregated $\sigma^{r+1}$ and $\psi^{r+1}$. For helper agents, server retrieves $H$ helper agents, $\psi^{h_{1:H}}$, to each client for every 10 rounds. More details of the training procedures for FedMatch, for the labels-at-client scenario, is described in Algorithm 1.

## 5 LABELS-AT-SERVER SCENARIO

**Problem Definition** We now describe another realistic setting, which is the labels-at-server scenario. This scenario assumes that the supervised labels are only available at the server, while local clients work with unlabeled data as described in Figure 1 (b). This is a common case of real-world applications where labeling requires expert knowledge (e.g. annotating medical images, evaluating body postures for exercises), but the data cannot be shared due to privacy concerns. In this scenario, $\mathcal{S}^G$ is identical to $\mathcal{S}$ and is located at server. The overall learning procedure is the same as that of conventional federated learning, except the global model $G$ performs supervised learning on $\mathcal{S}^G$ by minimizing the loss $\ell_s(\boldsymbol{\theta}^G)$ before broadcasting $\boldsymbol{\theta}^G$ to local clients. Then, the active local clients $l_{1:A}$ at communication round $r$ perform unsupervised learning which solely minimizes $\ell_u(\boldsymbol{\theta}^{l_a})$ on the unlabeled data $\mathcal{U}^{l_a}$.

**Algorithm 2 Labels-at-Server Scenario**

---

1: **RunServer()**
2: initialize $\sigma^0, \psi^0$
3: **for** each round $r = 1, 2, ..., R$ **do**
4:     **for** each server epoch $e$ from 1 to $E_G$ **do**
5:         **for** minibatch $s \in \mathcal{S}_G$ **do**
6:             $\theta_{\sigma+\psi^*} \leftarrow \theta_{\sigma+\psi^*} - \eta \nabla \ell_s(\theta_{\sigma+\psi^*}; s)$
7:         **end for**
8:     **end for**
9:     $\mathcal{L}^r \leftarrow$ (select random $A$ clients from $\mathcal{L}$)
10:    **for** each client $l_a^r \in \mathcal{L}^r$ **in parallel do**
11:       $\psi_{1:H}^r \leftarrow$ GetNearestNeighbors$(\psi^r)$
12:       $\psi_a^r \leftarrow$ RunClient$(\sigma^{r+1}, \psi^r, \psi_{1:H}^r)$
13:       EmbedLocalModel$(\sigma^{r+1}, \psi_a^r)$
14:     **end for**
15:    $\psi^{r+1} \leftarrow \frac{1}{A} \sum_{a=1}^{A} (\psi_{l_a}^r)$
16: **end for**
17: **RunClient($\sigma, \psi, \psi_{1:H}$)**
18: $\theta_l \leftarrow \sigma^* + \psi, \theta_{h_{1:H}} \leftarrow \sigma^* + \psi_{1:H}$
19: **for** each local epoch $e$ from 1 to $E_L$ **do**
20:     **for** minibatch $u \in \mathcal{U}_{l_a}$ **do**
21:         $\theta_{\sigma^*+\psi} \leftarrow \theta_{\sigma^*+\psi} - \eta \nabla \ell_u(\theta_{\sigma^*+\psi}; \theta_{h_{1:H}}, u)$
22:     **end for**
23: **end for**

Figure 4: **Illustrative Running Example of Labels-at-Server Scenario** We depict learning and transmitting procedure between a client and the global server under Labels-at-Server scenario corresponding to the Algorithm 2. Note that, in labels-at-server scenario, the labeled data is only available at the server, and thus global model at the server learns on labeled data, while local models at clients learn on only unlabeled data. Further details are explained in Section 5.

**FedMatch Algorithms for Labels-at-Server Scenario** We now describe our *FedMatch* algorithm for the labels-at-server scenario. As depicted in Figure 4, which describes an illustrative running example for labels-at-server scenario, the global model $G$ learns $\sigma$ on labeled data $\mathcal{S}^G$ at the server and the active local clients $l_{1:A}$ at the current round $r$ learn $\psi^{l_{1:A}}$ on unlabeled data $\mathcal{U}^{1:A}$ at each local environment. After the completion of local training, clients update their learned knowledge $\psi^{l_{1:A}}$ to the server. The server then embeds local models based on model similarity and create a KD-Tree for rapid nearest neighbor search for the top-$H$ most similar $\psi^{h_{1:H}}$ models for each client. At the next round, server transmits its learned $\sigma^{r+1}$ and the aggregated $\psi^{r+1}$. Server transmits top-$H$ similar $\psi^{h_{1:H}}$ to each client for every 10 communication rounds. Further training details of FedMatch for the labels-at-server scenario is described in Algorithm 2.

# 6 EXPERIMENTS

We now experimentally validate our method, *FedMatch*, on three tasks, such as Batch-IID, Batch-NonIID, and Streaming-NonIID, under both scenarios, *Labels-at-Client* and *Labels-at-Server*.

## 6.1 EXPERIMENTAL SETUP

**Tasks** **1) Batch-IID**: We use CIFAR-10 for this task and split $60,000$ instances into training $(54,000)$, valid $(3,000)$, and test $(3,000)$ sets. We extract 5 labeled instances per class ($C$=10) for each client ($K$=100) as labeled set $\mathcal{S}$, and the rest of instances $(49,000)$ are used as unlabeled data $\mathcal{U}$, so that we can evenly split $\mathcal{S}$ and $\mathcal{U}$ into $\mathcal{S}^{l_{1:100}}$ and $\mathcal{U}^{l_{1:100}}$, such that local models $l_{1:100}$ learn on corresponding labeled and unlabeled data during training. **2) Batch-NonIID (class-imbalanced)**: The setting of this task is mostly the same with the Batch-IID task, except we arbitrarily control the distribution of the number of instances per class for each client to simulate *class-imbalanced* environments. **3) Streaming-NonIID (class-imbalanced)**: In this task, data streams into each client from class-imbalanced distributions. We use Fashion-MNIST dataset for this task, and split $70,000$ instances into training $(63,000)$, valid $(3,500)$, and test $(3,500)$ sets. From train set, we extract 5 labeled instances per class ($C$=5) for each client ($K$=10) for a labeled set $\mathcal{S}$. We discard labels for the rest of instances to construct an unlabeled set $\mathcal{U}$ $(62,000)$. Then, we split $\mathcal{S}$ and $\mathcal{U}$ into $\mathcal{S}^{l_{1:100}}$ and $\mathcal{U}^{l_{1:100}}$ based on a class-imbalanced distribution. For individual local unlabeled data $\mathcal{U}^{l_k}$, we again split all instances into $\mathcal{U}_t^{l_k}$, $t \in \{1, 2, ..., T\}$, where $T$ is the number of total streaming steps (we set $T$=10). We train each streaming step for 10 rounds. We describe above tasks under Labels-at-Client scenario. For Labels-at-Server scenario, $S$ is simply located at server without any partition. Please see Figure 7 in Appendix, which we visualize the concepts of dataset configuration.

Table 1: **Performance Comparison on Batch-IID & NonIID Tasks** We use 100 clients ($F$=0.05) for 200 rounds. We measure global model accuracy and averaged communication costs. Note that the SL (Supervised Learning) models learn on both $\mathcal{S}$ and $\mathcal{U}$ with full labels, and are utilized as the upper bounds for each experiment.

| **CIFAR-10, Batch-IID Task with 100 Clients** ($K$=100, $F$=0.05, $H$=2) | | | | | | |
|---|---|---|---|---|---|---|
| | *Labels-at-Client* Scenario | | | *Labels-at-Server* Scenario | | |
| **Methods** | **Acc.(%)** | **S2C Cost** | **C2S Cost** | **Acc.(%)** | **S2C Cost** | **C2S Cost** |
| FedAvg-SL | $58.60 \pm 0.42$ | 100 % | 100 % | $52.45 \pm 0.23$ | 100 % | 100 % |
| FedProx-SL | $59.30 \pm 0.31$ | 100 % | 100 % | $49.11 \pm 0.38$ | 100 % | 100 % |
| FedAvg-UDA | $46.35 \pm 0.29$ | 100 % | 100 % | $24.81 \pm 0.73$ | 100 % | 100 % |
| FedProx-UDA | $47.45 \pm 0.21$ | 100 % | 100 % | $19.91 \pm 0.31$ | 100 % | 100 % |
| FedAvg-FixMatch | $47.01 \pm 0.43$ | 100 % | 100 % | $11.95 \pm 0.60$ | 100 % | 100 % |
| FedProx-FixMatch | $47.20 \pm 0.12$ | 100 % | 100 % | $25.61 \pm 0.32$ | 100 % | 100 % |
| **FedMatch (Ours)** | $52.13 \pm 0.34$ | 79 % | 46 % | $44.95 \pm 0.49$ | 45 % | 22 % |
| **CIFAR-10, Batch-NonIID Task with 100 Clients** ($K$=100, $F$=0.05, $H$=2) | | | | | | |
| FedAvg-SL | $55.15 \pm 0.21$ | 100 % | 100 % | $51.50 \pm 0.51$ | 100 % | 100 % |
| FedProx-SL | $57.75 \pm 0.15$ | 100 % | 100 % | $49.31 \pm 0.18$ | 100 % | 100 % |
| FedAvg-UDA | $44.35 \pm 0.39$ | 100 % | 100 % | $27.61 \pm 0.71$ | 100 % | 100 % |
| FedProx-UDA | $46.31 \pm 0.63$ | 100 % | 100 % | $26.01 \pm 0.78$ | 100 % | 100 % |
| FedAvg-FixMatch | $46.20 \pm 0.52$ | 100 % | 100 % | $09.45 \pm 0.34$ | 100 % | 100 % |
| FedProx-FixMatch | $45.55 \pm 0.63$ | 100 % | 100 % | $09.21 \pm 0.24$ | 100 % | 100 % |
| **FedMatch (Ours)** | $52.25 \pm 0.81$ | 85 % | 49 % | $44.17 \pm 0.19$ | 42 % | 20 % |

(a) Labels-at-Client Scenario        (b) Labels-at-Server Scenario

Figure 5: **Test Accuracy Curves on Batch-IID & NonIID Tasks** We visualize test accuracy curves of model performance corresponding to the Table 1. Note that the SL (Supervised Learning) models learn on both $\mathcal{S}$ and $\mathcal{U}$ with full labels, and are utilized as the upper bounds for each experiment.

**Baselines and Training Details**   Our baselines are: **1) Local-SL**: local supervised learning (SL) with full labels ($\mathcal{S} + \mathcal{U}$) without sharing locally learned knowledge. **2) Local-UDA** and **3) Local-FixMatch**: local semi-supervised learning, including UDA and FixMatch, without sharing local knowledge. **4) FedAVG-SL** and **5) FedProx-SL**: supervised learning with full labels ($\mathcal{S} + \mathcal{U}$) while sharing local knowledge via FedAvg and FedProx frameworks. **6) FedAvg-UDA** and **7) FedProx-UDA**: naive combinations of FedAvg/Prox with UDA. **8) FedAvg-FixMatch** and **9) FedProx-FixMatch**: naive combination of with FixMatch with FedAvg/Prox. For training, we use SGD with adaptive-learning rate decay introduced in (Serra et al., 2018) with the initial learning rate $1e-3$. We use **ResNet-9** networks as the backbone architecture for all baselines and our methods. We ensure that all hyper-parameters are set equally for all base models and ours to perform fair evaluation and comparison. Please see the **Section A** in the Appendix for further details. For all experiments, we report the mean and the standard deviation over 3 runs.

## 6.2 EXPERIMENTAL RESULTS

**Results on Batch-IID & NonIID Tasks**   Table 1 shows performance comparison of our models and naive Fed-SSL algorithms on Batch-IID and NonIID tasks under the two different scenarios. We observe that our model outperforms all naive Fed-SSL baselines for all tasks and scenarios. In particular, under labels-at-server scenario, which is more challenging than labels-at-client scenario, we observe that the naive combination models significantly suffer from the forgetting issue and their performances keeps deteriorating after a certain communication round. This phenomenon is mainly caused by the base models failing to properly perform disjoint learning, in which case the learned knowledge from the labeled and unlabeled data causes inter-task interference. Contrarily, our methods show consistent and robust performance regardless where the labeled data exists, which shows that our decomposition techniques effectively handles the challenging disjoint learning scenario. In addition, when the class-wise distribution is imbalanced for each client (Non-IID task), we observe that the base models' performance slightly drops by 1-3%$p$, while our methods show consistent.

Table 2: **Averaged Local Performance on Streaming-NonIID Task** We use 10 clients 100 rounds. We measure averaged local model accuracy and communication costs. Note that the SL models learn on both $\mathcal{S}$ and $\mathcal{U}$ with full labels, and are utilized as the upper bounds for each experiment.

| **Fashion-MNIST, Streaming-NonIID Task with 10 Clients** ($K$=10, $F$=1.0, $H$=2) | | | | | | |
|---|---|---|---|---|---|---|
| | *Labels-at-Cleint* Scenario | | | *Labels-at-Server* Scenario | | |
| **Methods** | **Acc.(%)** | **S2C Cost** | **C2S Cost** | **Acc.(%)** | **S2C Cost** | **C2S Cost** |
| Local-SL | $87.19 \pm 0.36$ | N/A | N/A | N/A | N/A | N/A |
| Local-UDA | $70.70 \pm 0.28$ | N/A | N/A | N/A | N/A | N/A |
| Local-FixMatch | $62.62 \pm 0.32$ | N/A | N/A | N/A | N/A | N/A |
| FedProx-SL | $82.06 \pm 0.26$ | 100 % | 100 % | $77.43 \pm 0.42$ | 100 % | 100 % |
| FedProx-UDA | $73.71 \pm 0.17$ | 100 % | 100 % | $83.34 \pm 0.21$ | 100 % | 100 % |
| FedProx-FixMatch | $62.40 \pm 0.43$ | 100 % | 100 % | $73.71 \pm 0.32$ | 100 % | 100 % |
| **FedMatch (Ours)** | $\mathbf{77.95 \pm 0.14}$ | **37 %** | **48 %** | $\mathbf{84.15 \pm 0.31}$ | **14 %** | **63 %** |

(a) Inter-Client Consistency     (b) Sigma & Psi     (c) Inter-Task Interference     (d) Number of Labels

Figure 6: **Ablation Study and Additional Analysis on FedMatch Algorithm** We study effectiveness of each components of our method, (a) *inter-client consistency loss* and (b) *parameter decomposition*. (c) We effectively tackle the *inter-task interference*. (d) Performance improvement of our method when labeled data is increased.

This shows that our inter-client consistency effectively enhances consistency with the helper agents selected from server based on model similarity, which is good at, in particular, class-imbalanced tasks. We also visualize the test accuracy curve for our models and naive Fed-SSL in Fig. 5. Our method (Red line) trains faster and consistently outperforms the base models, and is most robustness against inter-task interference in both scenarios. For analysis on the averaged communication costs, please see Section B.1 in the Appendix.

**Results on Streaming-NonIID Task**   Table 2 shows averaged local model performance on Streaming-NonIID tasks with 10 synchronized clients. For the labels-at-client scenario, our proposed method outperforms local-SSL and naive Fed-SSL models with large margins, $4\text{-}15\%p$, except for the SL models. There is no huge difference of performance between local SSL and Fed-SSL models, and this implies that our method effectively utilizes inter-client knowledge in this streaming setting. In the labels-at-server scenario, interestingly, the performance of FedProx-SL decreases by around $5\%p$ compared to the labels-at-client scenario, while Fed-SSL models obtain improved performance. We conjecture that this is because, for streaming situation, the model may not sufficiently train on the new data, while Fed-SSL models overcomes it by utilizing only the consistent pseudo-labels. Even on this task, FedMatch outperforms all baselines with significantly smaller communication cost on average (see Section B.1 for detailed analysis of the averaged communication costs).

**Effectiveness of Inter-Client Consistency**   To show the effectiveness of our inter-client consistency loss, we eliminate the loss, while learning on Batch-IID task with 100 clients ($F$=0.05). In Figure 6 (a), when we remove our inter-client consistency loss, we observe that the performance has slightly dropped (Pink line) from one with the loss term (Red line). This gap clearly tells us that our inter-client consistency loss improves model consistency across multiple models while keeping reliable knowledge. Interestingly, our model without inter-client consistency loss still outperforms base models. This additionally implies that our another proposed method, parameter decomposition for disjoint learning, also effectively enhances model performance.

**Effectiveness of Parameter Decomposition**   Our model with parameter decomposition alone, without inter-client consistency loss, outperforms base models. For further analysis, we show the effect of each decomposed variables, $\sigma$ and $\psi$, in Figure 6 (b). Removing either of $\sigma$ and $\psi$ results in substantial drop in the performance, with larger performance degeneration when dropping $\sigma$, which captures much more essential knowledge from labeled data (Green line). Such decomposition is effective since there exists knowledge interference between supervised learning and unsupervised learning. We show this with an experiment where we perform semi-supervised learning with 5 labeled instances per class and $1,000$ unlabeled instances for 100 rounds. We measure accuracy on the labeled set at each training steps. As shown in Figure 6 (c), our method effectively preserves

learned knowledge from labeled set, while other base models suffer from knowledge interference. This effective separation of supervised and unsupervised learning tasks enhances the overall performance of our methods even without inter-client consistency loss as shown in Figure 6 (a) (shown in pink).

**Number of Labels per Class**    We increase the number of labels per class for each client in a range of 1, 5, 10, and 20 on Batch-IID task (CIFAR-10). Our method shows consistent performance improvement as the number of labels increases. Interestingly, we observe that baseline models, FedProx-UDA/FixMatch, show performance degradation even when the labeled data increases ($5 \rightarrow 10$). These results show that our method effectively utilize knowledge from labeled and unlabeled data in federated semi-supervised learning settings, while other naive combinations of FSSL could fail to learn properly from labeled and unlabeled data in federated learning framework.

## 7    RELATED WORK

**Federated Learning**    A variety of approaches for averaging local weights at server have been introduced in the past few years. FedAvg (McMahan et al., 2017) performs weighted-averaging on local weights according to the local train size. FedProx (Li et al., 2018) uniformly averages the local updates while clients perform proximal regularization against the global weights, while FedMA (Wang et al., 2020) matches the hidden elements with similar feature extraction signatures in layer-wise manner when averaging local weights. PFNM (Yurochkin et al., 2019) introduces aggregation policy which leverages Bayesian non-parametric methods. Beyond focusing on averaging local knowledge, there are various efforts to extend FL to the other areas, such as continual learning under federated learning frameworks (Yoon et al., 2020a) inspired by parameter decomposition techniques proposed by (Yoon et al., 2020b). Recently, interests of tackling scarcity of labeled data in FL are emerging and discussed in (Jin et al., 2020; Guha et al., 2019; Albaseer et al., 2020).

**Semi-Supervised Learning**    While there exist numerous work on SSL, we mainly discuss consistency regularization approaches. Consistency regularization (Sajjadi et al., 2016) assumes that the class semantics will not be affected by transformations of the input instances, and enforces the model output to be the same across different input perturbations. Some extensions to this technique perturb inputs adversarially (Miyato et al., 2018), through dropout (Srivastava et al., 2014), or through data augmentation (French et al., 2018). UDA (Xie et al., 2019) and ReMixMatch (Berthelot et al., 2019a) use two sets of augmentations, weak and strong, and enforce consistency between the weakly and strongly augmented examples. Recently, in addition to enforcing consistency between weak-strong augmented pairs, FixMatch (Sohn et al., 2020) performs pseudo-label refinement on model predictions via thresholding. Entropy minimization (Grandvalet & Bengio, 2004) which enforces the classifier to predict low-entropy on unlabeled data, is another popular technique for SSL. Pseudo-Label (Lee, 2013) constructs one-hot labels from highly confident predictions on unlabeled data and uses these as training targets inn a standard cross-entropy loss. MixMatch (Berthelot et al., 2019c) performs sharpening on target distribution on unlabeled data, to further refine the generated pseudo-label.

## 8    CONCLUSION

In this work, we introduced two practical scenarios of *Federated Semi-Supervised Learning* (FSSL) where each client learns with only partly labeled data (*Labels-at-Client* scenario), or supervised labels are only available at the server, while clients work with completely unlabeled data (*Labels-at-Server* scenario). To tackle the problem, we propose a novel method, *Federated Matching* (FedMatch), which introduces the *inter-client consistency loss* that aims to maximize the agreement between the models trained at different clients, and the *parameter decomposition for disjoint learning* which decomposes the parameters into one for labeled data and the other for unlabeled data for preservation of reliable knowledge, reduction of communication costs, and disjoint learning. Through extensive experimental validation, we show that FedMatch significantly outperforms both local semi-supervised learning methods and naive combinations of federated learning algorithms with semi-supervised learning on diverse and realistic scenarios. As future work, we plan to further improve our model to tackle the scenario where pretrained models deployed at each client adapts to a completely unlabeled data stream (e.g. on-device learning of smart speakers).

**Acknowledgements**   This work was supported by Samsung Research Funding Center of Samsung Electronics (No. SRFC-IT1502-51), Samsung Advanced Institute of Technology, Samsung Electronics Co., Ltd., Next-Generation Information Computing Development Program through the National Research Foundation of Korea(NRF) funded by the Ministry of Science, ICT & Future Plannig (No. 2016M3C4A7952634), the National Research Foundation of Korea(NRF) grant funded by the Korea government(MSIT) (2018R1A5A1059921), and Center for Applied Research in Artificial Intelligence (CARAI) grant funded by DAPA and ADD (UDI190031RD). Also, this work was supported by Institute of Information  communications Technology Planning  Evaluation (IITP) grant funded by the Korea government(MSIT) (No.2019-0-00075, Artificial Intelligence Graduate School Program(KAIST))

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

**Organization**   We describe detailed experimental setups in Section A, such as our baselines (Section A.1), model architecture (Section A.2), and the training configurations (Section A.3). We also provide additional analysis and experimental results in Section B, including analysis on communication costs (Section B.1) and number of labels per class (Section B.2), experiments on real-world dataset (Section B.3), different backbone architecture (Section B.4), fraction of clients per communication round (Section B.5).

# A   EXPERIMENTAL DETAILS

We describe our experimental setups in detail, such as our baseline models, network architecture that is used for all base models and our method, and the detailed training setups.

## A.1   BASELINE MODELS

We consider UDA (Xie et al., 2019) and FixMatch (Sohn et al., 2020) as our baselines, since they are state-of-the-art SSL models and are based on the consistency-based mechanisms that are conceptually similar to our inter-client consistency loss. We reimplement UDA with the Training Signal Annealing (TSA) and exponential scheduling for its best performance as reported in their paper (we use RandAugment (Cubuk et al., 2019) for consistency regularization with random magnitude). We also reimplement FixMatch algorithms with strong augmentation as RandAugment (Cubuk et al., 2019). For weak augmentation (filp-and-shift), however, as the performance has significantly dropped when we apply the weak augmentation, we use original images rather than weakly augmenting the images. We fix confidence threshold $\tau$=0.85 for all FixMatch and our model experiments. For federated learning frameworks, we use FedAvg (McMahan et al., 2017) and FedProx (Li et al., 2018) algorithms since they are the standard baselines for federated learning and can be easily combined with the SSL baselines. Detailed hyper-parameter settings are described in Table 4.

## A.2   NETWORK ARCHITECTURE

We build ResNet-9 networks as our base architecture for all base model and our method. In the architecture, the first two convolutional neural layers have 64 and 128 filters and the same $3 \times 3$ kernel sizes followed by $2 \times 2$ max-pooling layer. Then we have a skip connection between the subsequent two convolution layers with 128 filters. We then double the filter size from 128 to 256 with the next conv layer and down-sample via the following $2 \times 2$ max-pooling layer. We repeat the previous step, such that we have 512 filter size and $4 \times 4$ kernel size. Then, we perform another skip connec-

Table 3: **Network Architecture of ResNet-9**

| Layer | Filter Shape | Stride | Output |
|---|---|---|---|
| Input | N/A | N/A | $32 \times 32 \times 3$ |
| Conv 1 | $3 \times 3 \times 3 \times 64$ | 1 | $32 \times 32 \times 64$ |
| Conv 2 | $3 \times 3 \times 64 \times 128$ | 1 | $32 \times 32 \times 128$ |
| Pool 1 | $2 \times 2$ | 2 | $16 \times 16 \times 128$ |
| Conv 3 | $3 \times 3 \times 128 \times 128$ | 1 | $16 \times 16 \times 128$ |
| Conv 4 | $3 \times 3 \times 128 \times 128$ | 1 | $16 \times 16 \times 128$ |
| Conv 5 | $3 \times 3 \times 128 \times 256$ | 1 | $16 \times 16 \times 256$ |
| Pool 2 | $2 \times 2$ | 2 | $8 \times 8 \times 256$ |
| Conv 6 | $3 \times 3 \times 256 \times 512$ | 1 | $8 \times 8 \times 512$ |
| Pool 3 | $2 \times 2$ | 2 | $4 \times 4 \times 512$ |
| Conv 7 | $3 \times 3 \times 512 \times 512$ | 1 | $4 \times 4 \times 512$ |
| Conv 8 | $3 \times 3 \times 512 \times 512$ | 1 | $4 \times 4 \times 512$ |
| Pool 4 | $4 \times 4$ | 4 | $1 \times 1 \times 512$ |
| Softmax | $512 \times 10$ | N/A | $1 \times 1 \times 10$ |

tion through the two subsequent conv layers with 512 filters. As a final step, we down-sample the kernel size from $4 \times 4$ to $1 \times 1$, then perform softmax classifier with the last fully connected layer. All layers are equally initialized based on the varaiance scalining method. The model architecture is described in Table 3.

## A.3   TRAINING DETAILS

We use Stochastic Gradient Descent (SGD) to optimize our model with initial learning rate 1e-3. We also adopt adaptive learning rate decay which is introduced by (Serra et al., 2018). The learning rate strategy gradually reduces the learning rate by a factor of 3 for every 5 epochs that validation loss does not consecutively decreases. We use L2 weight decay regularization on the base architecture with L2 factor to be 1e-4. All hyper-parameters and other training setups are equally set for fair comparison as shown in Table 4. In the table, we denote LPC as number of labels per class for each

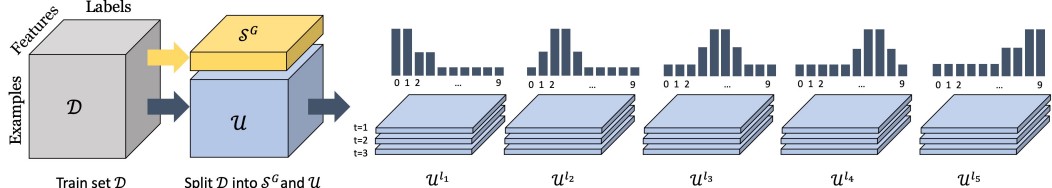

Figure 7: **Illustration of Dataset Partition for Experimental Tasks** We split the dataset $\mathcal{D}$ into a set of labeled data $\mathcal{S}$ and a set of unlabeled data $\mathcal{U}$. $U$ is divided into $K$ subsets which are distributed to $K$ clients (Batch Task). For *streaming* tasks, we further split all instances in each subset into $T$ subsets for $T$ streaming steps. For *class-imbalanced* tasks, we additionally control the number of instances per class for each client.

Table 4: **Hyper-Parameters & Training Setups** We provide all hyper-parameters and training setups for all baseline models and our method. Detailed hyper-parameters are also available in the code.

| Methods | lr | wd | $\lambda_s$ | $\lambda_u$ | $\lambda_{\text{ICCS}}$ | $\lambda_{\text{L}_1}$ | $\lambda_{\text{L}_2}$ | LPC | $B^{\mathcal{S}}_{\text{client}}$ | $B^{\mathcal{U}}_{\text{client}}$ | $B^{\mathcal{S}}_{\text{server}}$ | $\mu$ |
|---|---|---|---|---|---|---|---|---|---|---|---|---|
| **Labels-at-Client Scenario** | | | | | | | | | | | | |
| SL | 1e-3 | 1e-4 | 10 | - | - | - | - | 5 | 10 | 100 | - | 1e-2 |
| UDA | 1e-3 | 1e-4 | 10 | 1 | - | - | - | 5 | 10 | 100 | - | 1e-2 |
| FixMatch | 1e-3 | 1e-4 | 10 | 1 | - | - | - | 5 | 10 | 100 | - | 1e-2 |
| FedMatch | 1e-3 | 1e-4 | 10 | - | 1e-2 | 1e-4 | 10 | 5 | 10 | 100 | - | - |
| **Labels-at-Server Scenario** | | | | | | | | | | | | |
| SL | 1e-3 | 1e-4 | 10 | - | - | - | - | 100 | - | 100 | 100 | 1e-2 |
| UDA | 1e-3 | 1e-4 | 10 | 1 | - | - | - | 100 | - | 100 | 100 | 1e-2 |
| FixMatch | 1e-3 | 1e-4 | 10 | 1 | - | - | - | 100 | - | 100 | 100 | 1e-2 |
| FedMatch | 1e-3 | 1e-4 | 10 | - | 1e-2 | 1e-5 | 10 | 100 | - | 100 | 100 | - |

client (or at server). $B^{\mathcal{S}}$ and $B^{\mathcal{U}}$ denote batch-size of labeled set $\mathcal{S}$ and unlabeled set $\mathcal{U}$. $\mu$ is a hyper-parameter for FedProx framework. We additionally provide visual illustration of our dataset configuration. Please see Figure 7.

# B  ADDITIONAL ANALYSIS AND EXPERIMENTAL RESULTS

In this section, we additionally provide more analysis and experimental results, such as analysis on communication costs and number of labels per class, experiments on real-world dataset, different backbone architecture, fraction of clients per communication round.

## B.1  THE EFFICIENT COMMUNICATION OF FEDMATCH

Since the actual bit-level compression techniques are rather implementation issues, which are beyond our research scope, we only consider the reduction of *the amount of information* that needs to be transmitted between the server and the client. To minimize the communication costs, we not only learn $\psi$ to be sparse, but also subtract the parameters between server and client, such that $\Delta\psi = \psi^l_r - \psi^G_r$ and $\Delta\sigma = \sigma^l_r - \sigma^G_r$, then send only the difference, $\Delta\psi$ and $\Delta\sigma$, as sparse matrices from both directions of server-to-client (S2C) and client-to-server (C2S). Here, S2C and C2S costs are the sums of $\Delta\sigma$ and $\Delta\psi$. When transmitting the difference for each parameter to either way, we discard almost unchanged values in an element-wise manner, so that only meaningful neural values can be updated either server- or client-side. We observe that the range of the threshold values is from 1e-5 to 5e-5, such that the model performance is well-preserved and not significantly harmed, while maximizing the reduction of communication costs.

As shown in Figure 8, we observe that both the S2C and C2S costs are gradually decreased during the learning phases on both batch and streaming datasets under labels-at-client (Figure 8 (a) and (b)) and labels-at-server scenarios (Figure 8 (c) and (d)). This is because each parameter separately learns different tasks (i.e. supervised and unsupervised learning) effectively, which results in rapid convergence to optimal points, respectively. Further, for the labels-at-server scenario, since labeled data is not available at client, client even does not need to transfer $\Delta\sigma$ to the server (see Figure 8 (c) right and (d) right), which is extremely efficient than the labels-at-client scenario where both $\Delta\sigma$ and $\Delta\psi$ must be transferred to the server. For both scenarios, indeed, S2C contains the cost

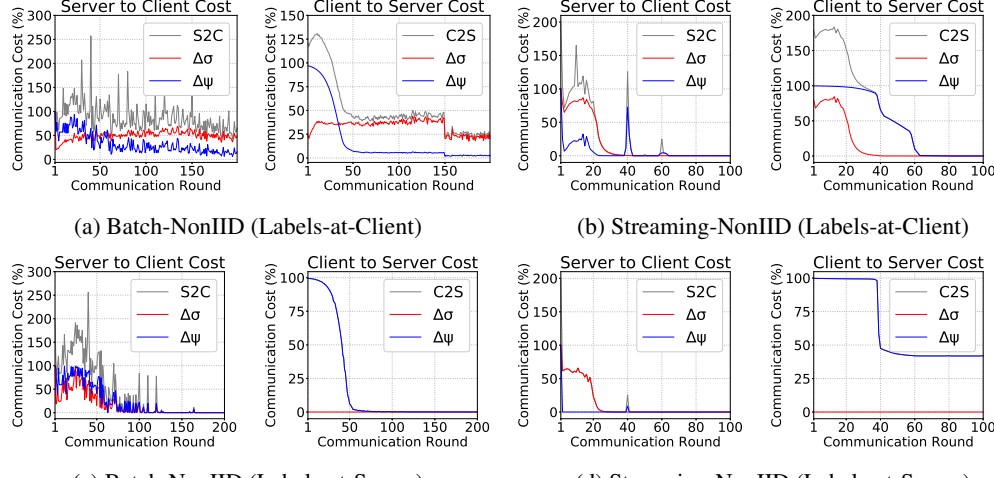

(a) Batch-NonIID (Labels-at-Client)   (b) Streaming-NonIID (Labels-at-Client)

(c) Batch-NonIID (Labels-at-Server)   (d) Streaming-NonIID (Labels-at-Server)

Figure 8: **Communication Cost Curves of FedMatch (ResNet-9) Corresponding to the Table 1 and 2**. We measure the communication costs for each parameters, $\Delta\sigma$ and $\Delta\psi$, during training phase. The communication costs under the labels-at-client scenario are visualized in (a) and (b) on the upper row. (c) and (d) on the lower row represent the communication costs under labels-at-server scenario.

| COVID-19 Radiography Dataset | | |
|---|---|---|
| | Labels-at-Client | Labels-at-Server |
| **Methods** | **Acc.(%)** | **Acc.(%)** |
| F.Prx-UDA | $74.24 \pm 0.25$ | $80.11 \pm 0.18$ |
| F.Prx-FixMtch | $70.02 \pm 0.28$ | $72.15 \pm 0.14$ |
| **FedMatch** | $\mathbf{78.67} \pm \mathbf{0.23}$ | $\mathbf{84.32} \pm \mathbf{0.11}$ |

(a) Labels-at-Client   (b) Labels-at-Server

Figure 9: **Experimental Results on COVID-19 Radiography Dataset. Left**: Performance comparison of our method (FedMatch) with the naive federated semi-supervised learning algorithms (FedProx-UDA/FixMatch). **Right**: Test accuracy curves corresponding to the left performance table. Our method trains stably and consistently outperforms all base models.

of helper agents, such that $\Delta\psi^{1:H} = \sum_{j=1}^{H} \psi_r^j - \psi_r^l$. However, as shown in Figure 8, transmitting multiple helper agents ($H=2$ in our experiments) does not significantly affect the total S2C costs thanks to our novel decomposition techniques as well as efficient subtracting method, such that model reconstruction can be possible without meaningful information loss at either server- or client-side.

## B.2 FURTHER ANALYSIS OF THE NUMBER OF LABELS PER CLASS

We explain our analysis of the number of labels per class under Section 6.2, and here we further provide additional experimental results. We conduct experiments with our method without the decomposition technique. As shown in Table 5, our method without decomposition technique (indicated as FedMatch (w/o)) shows not much performance improvement when the number of labels per class

Table 5: **Analysis of the Number of Labels per Class**

| | Number of Labeled Examples per Class | | | |
|---|---|---|---|---|
| - | 1 | 5 | 10 | 20 |
| **Methods** | **Acc.(%)** | **Acc.(%)** | **Acc.(%)** | **Acc.(%)** |
| FedPrx*UDA | 31.95 | 47.45 | 41.4 | 47.15 |
| FedPrx*FxMtch | 30.01 | 47.2 | 34.25 | 44.5 |
| FedMatch (w/o) | **37.7** | 47.51 | 51.15 | 62.7 |
| FedMatch | 37.65 | **54.5** | **60.65** | **66.1** |

increases from 5 to 10 (around $3.x\%p$) than from 1 to 5 (around $9.x\%p$) and from 10 to 20 ($10.x\%p$), which are the similar tendency with the baseline models in Table 5. However, with the decomposition technique, our method shows consistent performance improvement, which implies that our proposed technique has the effectiveness to handle inter-task interference and preserve reliable knowledge in the novel federated semi-supervised learning scenarios.

Table 6: **Performance Comaprison utilizing AlexNet-Like architecture** We use 100 clients for 100 rounds for streaming task and 200 rounds for batch tasks. We measure global model accuracy, while varying experimental settings (i.e. fraction of available clients and the accessibility of labeled data).

| Experiments based on AlexNet-Like Architecture | | | | | |
|---|---|---|---|---|---|
| | **Streaming-NonIID** ($F$=1.0) | | **Batch-IID** (Labels-at-Client) | | |
| | Labels-at-Client | Labels-at-Server | $F$=0.05 | $F$=0.10 | $F$=0.20 |
| **Methods** | **Acc.(%)** | **Acc.(%)** | **Acc.(%)** | **Acc.(%)** | **Acc.(%)** |
| FedAvg-SL | $68.20 \pm 0.29$ | $70.51 \pm 0.11$ | $47.23 \pm 0.31$ | $47.87 \pm 0.73$ | $48.73 \pm 0.15$ |
| FedProx-SL | $68.47 \pm 0.13$ | $70.55 \pm 0.72$ | $47.54 \pm 0.28$ | $48.01 \pm 0.17$ | $49.20 \pm 0.64$ |
| FedAvg-UDA | $52.25 \pm 0.04$ | $46.28 \pm 0.32$ | $35.27 \pm 0.29$ | $35.20 \pm 0.53$ | $36.21 \pm 0.12$ |
| FedProx-UDA | $52.84 \pm 0.15$ | $46.35 \pm 0.31$ | $34.94 \pm 0.46$ | $36.67 \pm 0.73$ | $35.80 \pm 0.43$ |
| FedAvg-FixMatch | $57.09 \pm 0.89$ | $52.67 \pm 0.78$ | $32.33 \pm 0.51$ | $36.27 \pm 0.33$ | $37.61 \pm 0.05$ |
| FedProx-FixMatch | $57.12 \pm 0.41$ | $51.51 \pm 0.32$ | $36.83 \pm 0.23$ | $36.37 \pm 0.39$ | $37.40 \pm 0.18$ |
| **FedMatch (Ours)** | $\mathbf{63.84} \pm \mathbf{0.18}$ | $\mathbf{59.12} \pm \mathbf{0.35}$ | $\mathbf{41.67} \pm \mathbf{0.32}$ | $\mathbf{41.97} \pm \mathbf{0.14}$ | $\mathbf{42.18} \pm \mathbf{0.27}$ |

### B.3    EXPERIMENTS ON REAL-WORLD DATASET

To show our method consistently work with real-world dataset, we further conduct experiment on COVID-19 Radiography Dataset (Chowdhury et al., 2020) which is a real-world dataset that consists of X-ray images from COVID and non-COVID patients. The COVID-19 dataset contains 219 X-ray images from the patients diagnosed of COVID-19, 1341 images from normal (healthy) patients, and 1341 images from patients diagnosed of viral pneumonia. We use 10 clients with a fraction of 1.0 (communication rate). We use 5 labeled examples per class for each client, leaving the rest of the image as unlabeled. We find this setting to be realistic as the datasets are, since we may not not have skilled radiologists that can fully label the X-ray images taken at the local hospitals. We compared our method against baselines which naively combine semi-supervised learning and federated learning models during training 100 rounds. As shown in the left table of Figure 9, our method consistently outperforms all base models with large margins (around $4\%p$-$10\%p$) in both scenarios. The test accuracy curves in Figure 9, we can see that our method trains faster than the base models and shows more stability during training. We believe that these additional experimental results further strengthen our paper.

### B.4    BACKBONE ARCHITECTURE

Most existing works on federated learning considers smaller networks since the focus is on-device learning of low-resource devices, and thus we utilize a smaller backbone networks than ResNet-9. To verify that our method also successfully works on the smaller & different architecture, we adopt AlexNet-Like (Serra et al., 2018), of which the first three layers are convolutional neural layers with $64$, $128$, and $256$ filters with the $4$, $3$, and $2$ kernel sizes followed by the two fully-connected layers of $2048$ units, while $2 \times 2$ max-pooling layers are followed after each convolutional layer. In Table 6, for both Streaming-NonIID and Batch-IID tasks, our methods still outperforms all naive Fed-SSL models with the similar tendency with that of the results based on ResNet-9. This shows that our methods can be applied to the smaller and different base networks, and still effectively utilize inter-client and reliable knowledge across multiple clients than naive algorithms.

### B.5    FRACTION OF AVAILABLE CLIENTS PER COMMUNICATION ROUND

To see the effect of participation rate of clients, we increase the fraction of available clients per communication round in a range of $0.05$, $0.10$, and $0.2$. This means, for every round, server can connect to the arbitrary 5, 10, and 20 available clients out of 100 clients, and perform distributed learning on each individual local data through each client and updates global knowledge by aggregating the locally-learned knowledge. The experimental results are shown in Table 6 Batch-IID. We observe that the performances of all models are slightly improved when the faction increases. This is natural that the more knowledge the client updates, the more the global performance is improved. We are not able to find any extraordinary phenomenon on the fraction of the number of clients per round.

