# OpenReview forum: "Federated Semi-Supervised Learning with Inter-Client Consistency & Disjoint Learning"
_ICLR.cc/2021/Conference — ICLR 2021 Poster_

### Official Review · AnonReviewer1 · 2020-10-27
**Interesting paper with two main weaknesses: poor organization & lack of a motivating, real-world application dataset.**

**Rating:** 6
**Confidence:** 3

**Review:**

AFTER READING THE AUTHORS' REPLY, I HAVE CHANGED MY RATING TO 6.

Even though authors introduce an interesting approach for a problem that has many potential practical applications, the paper suffers from two main weaknesses:
1) it is poorly organized, which makes it very hard to follow
2) it lacks a compelling, real-world dataset

In terms of paper organization, in this reviewer's opinion, it may be beneficial to decouple and present in a serial manner the solutions to the two main scenarios (i.e., labels at client vs server). The parallel presentation makes the story harder to follow, as the reader has to keep switching context from one scenario to the other. The very long captions of Figures 2 & 3 are not helpful, and Section 4 comes a bit out of nowhere, given that the pseudo-code for the two proposed algorithms only appears in the appendix.
A better approach would be to use the body of the paper for an illustrative running example and the pseudo-code of the algorithms, while relegating the details to the appendix.

In terms of the empirical validation, it is disappointing to see  that you are using synthetic datasets. The paper would greatly benefit from having at least one real world application domain in which the proposed approaches "move the needle." There are far too many papers in which a novel approach does great on synthetic data without impacting the state-of-the-art results on real-world domains.


Other:
- caption of Figure 1: the data is "available" rather than "given" to the local client (twice, under both "a)" and "b)")
- page 2: you make references to Figure 6 a & b, but you mean Figure 1 a & b
- page 3: in line 3 of the 1st paragraph of 3.1, you refer to "D" without defining it
- please spell-check & proof-read the paper:
    - "Cleint" --> "Client" in the header of Table 1;
    - "manly" --> "mainly" on page 6

---

> ### Author Response · Authors · 2020-11-15
> **Response to Reviewer #1**
>
> We thank you for your constructive comments. Please see the response to individual comments below:
>
> **(1) In terms of paper organization, in this reviewer's opinion, it may be beneficial to decouple and present in a serial manner the solutions to the two main scenarios (i.e., labels at client vs server). The parallel presentation makes the story harder to follow, as the reader has to keep switching context from one scenario to the other. A better approach would be to use the body of the paper for an illustrative running example and the pseudo-code of the algorithms, while relegating the details to the appendix.**
>
> - Thank you for your suggestions, and we have completely reorganized our paper, **sequentially introducing the two main scenarios as suggested**. We separated the descriptions of the two different scenarios into two subsequent sections (Section 4 and Section 5 of the revision), to minimize context-switching between them. We have included illustrative running examples as well as the pseudo-codes of the algorithms for each section, in the revision. We believe that the paper has a largely improved organization after revising it according to your comment.
>
> ---
>
> **(2) The very long captions of Figures 2 & 3 are not helpful.**
>
> - We have **split the Figure 3**into Figure 3 and Figure 4 in the revision, as introduce the two scenarios in two separate sections. Thus the length of the corresponding captions are also significantly reduced, which we believe is more readable than the ones in the previous version of the paper.
>
> ---
>
> **(3) Section 4 comes a bit out of nowhere, given that the pseudo-code for the two proposed algorithms only appears in the appendix.**
>
> - We have included the pseudo-codes of the algorithms in the Appendix due to page limit, but have included them back into the main paper (Algorithm 1 in page 5, Algorithm 2 in page 6).
>
> ---
>
> **(4) In terms of the empirical validation, it is disappointing to see that you are using synthetic datasets. The paper would greatly benefit from having at least one real world application domain in which the proposed approaches "move the needle." There are far too many papers in which a novel approach does great on synthetic data without impacting the state-of-the-art results on real-world domains.**
>
> - We agree with your point that validation on the real-world datasets is important. However, please understand that it is difficult for Academic researchers to secure publicly available real-world datasets. For this reason, the majority of federated learning algorithms validate on such synthetic datasets, and we believe that our evaluation is fair when compared with the existing works. Using publicly available data is also beneficial for reproducibility.
>
> - Nevertheless, we searched and found [**COVID-19 RADIOGRAPHY DATABASE**](https://www.kaggle.com/tawsifurrahman/covid19-radiography-database) which is a real-world dataset that consists of X-ray images from COVID and non-COVID patients. The COVID-19 dataset contains 219 X-ray images from the patients diagnosed of COVID-19, 1341 images from normal (healthy) patients, and 1341 images from patients diagnosed of viral pneumonia. We use 10 clients with a fraction of 1.0 (communication rate). We use 5 labeled examples per class for each client, leaving the rest of the image as unlabeled. We find this setting to be realistic as the datasets are, since we may not not have skilled radiologists that can fully label the X-ray images taken at the local hospitals. We compared our method against baselines which naively combine semi-supervised learning and federated learning models during training 100 rounds. The results are as follow:
>
> |COVID-19 Radiography Dataset                              |||
> |:-------------------------:|:------------------------------:|:----------------------:|
> |         -                        |       Labels-at-Client       | Labels-at-Server |
> | Methods                 |                 Acc. (%)           |         Acc. (%)        |
> | FedProx-UDA         |           74.24 ± 0.25          |     80.11 ± 0.18     |
> | FedProx-FixMatch |           70.02 ± 0.28         |      72.15 ± 0.14     |
> | **FedMatch (Ours)**   |           **78.67 ± 0.23**         |      **84.32 ± 0.11**     |
>
> - As shown in the above table, our method consistently outperforms all base models with large margins (around 4%p-10%p) in both scenarios. We further provide test accuracy curves in Figure 8 of the Appendix. In the figure, our method trains faster than the base models and is more stable. We believe that these additional experimental results further strengthen our paper, and thank you for your suggestion.
>
> ---
>
> **(5) Others (typos)**
>
> - Thank you for the helpful suggestion. We have reflected all your comments regarding typos, incorrect references, caption descriptions, etc.) in the revision.

---

### Official Review · AnonReviewer2 · 2020-10-27
**problem description is complicated, not solving the original problem, contribution not clear**

**Rating:** 4
**Confidence:** 3

**Review:**

This paper studies the problem of learning a joint model for different local data sets. Each model is trained individually based on each individual local data set, and the inference is performed by regularizing the results of different models.  The main challenges come from the facts that each local data set belongs to different client hence the data is not shared and the labeled data in some local data set may be not sufficient. To utilize the unlabeled and labeled data, the paper proposes using unsupervised, semisupervised and supervised learning models. To incrementally training the model with different types of data, authors  propose to decompose the model parameters as supervised and unsupervised model. The description of the problem is complicated and little confusing.

The problem seems interesting, but one of my main main concerns is that I do not think authors solve the original challenge. The original challenge is that clients do not want to share data for model training. But the method proposed utilizes all the local data sets to learn the model.

My another main concern is that the novel contribution of this work is not strong. My impression of the method proposed in this work is pretraining the model with labeled data and fine-tuning it with unlabeled data with constraints to make the fine-tuned model as close as the pretrained model.

---

> ### Author Response · Authors · 2020-11-12
> **Response regarding "not solving the original challenge" and "technical novelty"**
>
> We sincerely thank you for your time and effort in reviewing our paper. We believe that your main concerns are based on misunderstandings of our paper. Please see the response to individual comments below:
>
> The problem seems interesting, but one of my main main concerns is that I do not think authors solve the original challenge. The original challenge is that clients do not want to share data for model training. But the method proposed utilizes all the local data sets to learn the model.
>
> - This is a critical misunderstanding since in our framework, the clients **do not share the local data** with the others at all, and thus there is no **compromise of data privacy**at the local clients. Could you elaborate why you thought this would be the case? What are shared instead in our framework, are the **parameters**learned on the local data, as with the conventional federated learning setting. Please see Figure 3 to see how the parameters are communicated across clients.
>
> ---
>
> My another main concern is that the novel contribution of this work is not strong.
>
> - We believe that you may have missed our contributions on the methodology part. Besides the unique setting we consider for federated semi-supervised learning ("Labels-at-Server" setting), we propose both **inter-client consistency loss**which makes the prediction across models at multiple clients to be consistent, and the **parameter decomposition**for disjoint learning on labeled and unlabeled data. Both are our unique technical contributions that can effectively solve the federated semi-supervised learning problem in the two practical scenarios we described. Please see the second bullet point in the contributions we list in Page 2, and Section 3.1 (Inter-client Consistency Loss, Parameter Decomposition for Disjoint Learning) for more detailed discussions of the contributions.
>
> ---
>
> My impression of the method proposed in this work is pretraining the model with labeled data and fine-tuning it with unlabeled data with constraints to make the fine-tuned model as close as the pretrained model.
>
> - This is another critical misunderstanding since our model **does not fine-tune the model pretrained** on the labeled data, on unlabeled data. The **fine-tuning approach you describe is rather done by the baseline federated learning models**(FedAvg*SL, FedProx*UDA, FedProx*FixMatch), which result in severe performance degeneration since they do not prevent the model from forgetting what they have learned about the labeled data (Please see Figure 5(b)). On the other hand, our method does not result in any performance degeneration as it keeps the training of the parameter for labeled data to be **completely disjoint**from the unlabeled one with parameter decomposition (Figure 5(b), FedMatch (Ours)), instead of finetuning the model learned on labeled data, with unlabeled data.
>
> - Thus there is **no pretrained model or fine-tuning** done with our model in neither of the two scenarios. In the Labels-at-Client scenario, both labeled and unlabeled data is generated locally, and the model learns on the two types of data concurrently. There is no fine-tuning happening in the Labels-at-Server scenario either, since the training for the labeled data and unlabeled data is also done at the same time Moreover, with our disjoint learning that allows to learn the parameters for the unlabeled data separately from the parameters for the labeled data, there is almost no interference across the learning for two different types of data  (Please see Figure 3 and Page 5).
>
>
>
> We hope that these comments clarify your concerns regarding the data privacy preservation and the technical novelty of our work. Please let us know if there are any other concerns.

---

> > ### Comment · AnonReviewer2 · 2020-11-16
> > **optimization of aggregating parameters not clear**
> >
> > It is hard to tell the data and neural network layers in Figure 3. The symbols such as $l_a$ in the narrative cannot be found in the figures. Since there are multiple local trained models, Figure 3 only shows the interaction of the server and local client.
> >
> > Since the main contribution of this work is to aggregate parameters of individually trained models, are the main objective functions Equation 4 and 5 all about training a local model with labeled and unlabeled data? How the parameters of different models  are aggregated is still not clear to me.

---

> > > ### Author Response · Authors · 2020-11-17
> > > **Response to Reviewer #2**
> > >
> > > Thank you for your response and constructive comments. Please see our response to the comments below:
> > >
> > > **(1-1) It is hard to tell the data and neural network layers in Figure 3.**
> > > - Please note that we have split Figure 3 into Figure 3 and 4 in the revision, for better organization and understanding. We have included the **visualization of the data**in **Figure 3 and 4**of the revision, as suggested.
> > >
> > > ---
> > >
> > > **(1-2) The symbols such as  $l_a$ in the narrative cannot be found in the figures.**
> > > - We have included the symbol $l_a$ in **Figure 3 and 4**of the revision.
> > >
> > > ---
> > >
> > > **(1-3) Although there are multiple locally trained models, Figure 3 only shows the interactions between the server and the local clients.**
> > >
> > > - Please note that there is only **a single local model (a neural network)** per client, whose parameters are decomposed into two sets of parameters, one for labeled data and the other for unlabeled data, at all layers. The inter-client consistency regularization in Figure 2 is enforced across models from different clients.
> > >
> > > - If you are referring to the parameters for the labeled data and unlabeled data at multiple layers of the neural network as "multiple models", then they are described in **Figure 3 and 4.** ($\sigma^{l_a}$ is the parameter labeled data and $\psi^{l_a}$ is the parameter for unlabeled data, where $l$ and $a$ is the client index)
> > >
> > > - Besides, we moved the pseudo-codes of the algorithms back to the main text for better clarity **(Algorithm 1 and 2)**. For detailed explanations, please see Section 4.2 and 5.2 of the **revised version of the paper** (We describe all symbols in Figure 3 and Figure 4).
> > >
> > > ---
> > >
> > > **(2-1) How the parameters of different models are aggregated is still not clear to me.**
> > >
> > > - How to aggregate parameters from multiple local models differs in the two scenarios we target.
> > >
> > > - **1) Labels-at-client scenario:** The server aggregates $\sigma$ and $\psi$ (respectively) across clients using FedAvg (or FedProx), since local models have both parameters to perform disjoint learning on labeled and unlabeled data.
> > >
> > > - **2) Labels-at-server scenario:** As the clients learn only unlabeled data at the local environment, the server aggregates only $\psi$ from multiple local models. The server does not need to aggregate $\sigma$, the parameter for the labeled data, in this case as it is only available and learned at the server.
> > >
> > > - For more detailed explanations, please see the Section 4.2 and Section 5.2 of the revision since we have included clear descriptions of how the parameters are aggregated for both scenarios. Please also refer to our algorithms for both scenarios in Algorithm 1 and 2.
> > >
> > > ---
> > >
> > > **(2-2) Since the main contribution of this work is to aggregate parameters of individually trained models, are the main objective functions Equation 4 and 5 all about training a local model with labeled and unlabeled data?**
> > >
> > > - Our main contribution is not just about aggregating parameters from the individual local models. We would like to recap our contributions, as we believe that our work is highly novel in multiple aspects:
> > >
> > > - **1) Novel problem:** We believe that our problem setting is highly novel, and is not a straightforward extension of semi-supervised learning in a federated learning setting, since "Labels-at-Server" scenario, where the labels are only available at the server, while clients only receive unlabeled data, is a unique but a realistic setting for federated learning.
> > >
> > > - **2) Inter-client consistency:** The inter-client consistency term is novel since existing approaches enforce consistency across predictions on the perturbations of the same sample (generated using data augmentation), while we enforce consistency across predictions from multiple models. Further, we do not straightforwardly enforce consistency across all models learned at all clients, since they may learn on heterogeneous data. Rather, we consider the task relevance to find the models that are helpful for a given task, by embedding the models based on their functional similarities and retrieving the nearest neighbors in the function space (Please see the last paragraph of Inter-Client Consistency part in the methodology).
> > >
> > > - **3) Disjoint learning:** To our knowledge, our is the first work that decomposes the parameters for labeled and unlabeled datafor semi-supervised learning, and this has effectively preventedthe unreliable knowledge from pseudo-labels from negatively affectingthe reliable knowledge captured with labeled data. We believe that this is an important contribution, since catastrophic forgetting of the knowledge of labeled data is more severe in federated learning scenarios, where the models train on sequential data rather than on batch data. The proposed disjoint learning also provides a natural solution to "Labels-at-Server" scenario, where we need to combine knowledge of the labeled data learned at the server, with the knowledge of the unlabeled data learned at the clients.

---

> > > > ### Comment · AnonReviewer2 · 2020-11-24
> > > > **formalize contribution with mathematical language**
> > > >
> > > > Thank you for your response.
> > > > Can you formalize your contributions using mathematical language?

---

> > > > > ### Author Response · Authors · 2020-11-25
> > > > > **Contributions formalized in mathematical language**
> > > > >
> > > > > Thank you for getting back to us. As you requested, we formalize our contributions, such as the introduction of a novel problem, and proposal of an inter-client consistency loss, and disjoint learning algorithms. Please also refer to **Section 2** (Problem Definition), **4** (Labels-at-Client), and **5** (Labels-at-Server), **3.1** (Inter-Client Consistency),  **3.2** (Disjoint Learning), in the revision document for more detailed descriptions.
> > > > >
> > > > > > **Federated Semi-Supervised Learning**: Let $G$ be a global model and $\mathcal{L}$ be a set of local models for $K$ clients. Let $\mathcal{D}$=${\\{\textbf{x}_i,\textbf{y}_i\\}}^{1:N}$ be a given dataset and $\mathcal{D}$ is split into a labeled set $\mathcal{S}$ and $K$ unlabeled sets $\mathcal{U}_1, \dots,\mathcal{U}_K$ that are privately located at $K$ clients. For a labeled set $\mathcal{S}$, we consider two different scenarios depending on the availability of labeled data at clients, namely the **Labels-at-Client** and the **Labels-at-Server** scenario. (Please see **Section 2** for more detailed descriptions)
> > > > >
> > > > > - **1) Labels-at-Client Scenario**: For this scenario, each client $k$ has both labeled dataset $\mathcal{S}^{k}$ and unlabeled data $\mathcal{U}^{k}$. At each round $r$, active local models $l_{1:A}$ perform semi-supervised learning by minimizing the loss  $\ell_{final}(\theta^{a})=\ell_{s}(\theta^{a})+\ell_{u}(\theta^{a})$ respectively on $\mathcal{S}^{a}$ and $\mathcal{U}^{a}$, where $a$ is the index of a selected local model. The global model $G$ aggregates updates from the selected subset of clients and broadcasts the aggregated knowledge to clients connected at the next round $r+1$. (Please refer to **Section 4** and **Figure 3** for detailed training algorithms and an illustrative running example)
> > > > >
> > > > > - **2) Labels-at-Server Scenario**: For this scenario, there is only one labeled dataset $\mathcal{S}^G$, located at the server. The global model $G$ performs supervised learning on $\mathcal{S}^G$ by minimizing the loss $\ell_{s}(\theta^{G})$ before broadcasting $\theta^G$ to local clients. Then, the active local clients $l_{1:A}$ at communication round $r$ perform unsupervised learning which solely minimizes $\ell_{u}(\theta^{a})$ on the unlabeled data $\mathcal{U}^{a}$. (Please refer to **Section 5** and **Figure 4** for detailed training algorithms and an illustrative running example)
> > > > >
> > > > > > **Inter-Client Consistency**: we propose a novel consistency loss which enforces the multiple models learned at multiple clients to output the same prediction as follows: $\Phi(\cdot)=\textrm{CrossEntropy}(\hat{\textbf{y}}, p_\theta(\textbf{y}|\pi(\textbf{u}))) +\frac{1}{H} \sum_{j=1}^H \textrm{KL}[p_{\theta^{\textrm{h}_j}}^*(\textbf{y}|\textbf{u})||p_\theta(\textbf{y}|\textbf{u})]$ (**Eq. (2) in the paper**), where $p^*_{\theta^{h}}(\textbf{y}|\textbf{x})$ are helper agents and $\pi(\textbf{u})$ performs RandAugment on an unlabeled instance $\textbf{u}$. The pseudo-label $\hat{\textbf{y}}$ is obtained by our novel agreement-based pseudo-labeling technique which produces one-hot labels on the class that has the maximum agreements. (Please see **Section 3.1** for in-depth explanation about inter-client consistency)
> > > > >
> > > > > > **Parameter Decomposition and Disjoint Learning**: we perform an **additive decomposition**of our model parameters $\theta$ into two variables, $\sigma$ for supervised learning and $\psi$ for unsupervised learning, such that $\theta=\sigma+\psi$. Note that while we use the **additively combined parameter $\theta$** for training with both labeled and unlabeled data, we only train one of the two parameters ($\sigma$ for labeled data and $\psi$ for unlabeled data) during training, while keeping the other parameter fixed.
> > > > >
> > > > > - On **labeled data**, we perform **supervised learning on $\sigma$**, while **keeping $\psi$ fixed**, by minimizing the loss term as follows: $\textrm{minimize}~\mathcal{L}_s(\sigma) = \lambda_s \textrm{CrossEntropy}(\textbf{y}, p_{\sigma+\psi^*}(\textbf{y}|\textbf{x}))$ (**Eq. (4) in the paper**), where $\textbf{x}$ and $\textbf{y}$ are from the labeled set $\mathcal{S}$.
> > > > >
> > > > > - On **unlabeled data**, we perform **unsupervised learning on $\psi$**, while **keeping $\sigma$ fixed**, by minimizing the consistency loss terms as follows: $\textrm{minimize}~\mathcal{L}_u(\psi) = \lambda_\textrm{ICCS}\Phi_{\sigma^*+\psi}(\cdot) +\lambda_2||\sigma^*-\psi||_2^2 +\lambda_1||\psi||_1 $ (**Eq. (5) in the paper**), where all $\lambda_1$ and $\lambda_2$ are hyper-parameters to control the learning ratio between the terms. (Please see **Section 3.2** for in-depth explanations)
> > > > >
> > > > > We hope this response clarifies your concerns regarding our contribution.

---

> > > ### Author Response · Authors · 2020-11-23
> > > **Reminder**
> > >
> > > Dear Reviewer,
> > >
> > > Could you please go over our responses and the revision since we can have interactions with you only by this Tuesday (24th)? We have responded to your comments and faithfully reflected them in the revision, and provided additional experimental results that you have requested. We sincerely thank you for your time and efforts in reviewing our paper, and your insightful and constructive comments.
> > >
> > > Thanks, Authors

---

> > > ### Author Response · Authors · 2020-11-24
> > > **The interactive discussion period will end soon.**
> > >
> > > Dear Reviewer,
> > >
> > > We have done our best to respond to your comments regarding clarity by providing you detailed responses, and revising the paper (Figure 3 and 4, Algorithm 1 and 2), in the revision. We also provided a summary of the contribution of our work, in the response. Could you please check the responses and the revision to see if they satisfactorily address your concerns since the interactive discussion period will end soon? We sincerely thank you for your helpful suggestions, which we strongly believe have further improved the quality of our paper.
> > >
> > > Best regards,
> > > Authors.

---

### Official Review · AnonReviewer3 · 2020-10-29
**Review comments for FSSL**

**Rating:** 6
**Confidence:** 3

**Review:**

In many real-world applications, local data are not always well labeled or fully labeled, and most of them are unlabeled. This paper introduced a novel learning paradigm, Federated Semi-Supervised Learning (FSSL), to handle the federated learning from distributed and partially labeled data within the clients. Under this paradigm, the paper studies two different scenarios where partly labeled data appear on client nodes and merely appear on the sever node. In the introduction, the authors clearly presented the motivations of introducing FSSL and showed the differences between the proposed method and some existing ones.

In the FSSL, the key technique is Federated Matching (FedMath), which learns inter-client consistency between clients, and decomposes parameters to reduce both interference between supervised and unsupervised tasks, and communication cost. The Inter-Client Consistency Loss was most directly taken from some existing method. Thus, the critical innovation is the Decomposition of parameters, which has several benefits to the learned models. Basically, the proposed method is intuitively and technically sound.

The authors conducted a batch of experiments to evaluate their proposed method on three different tasks under both labels-at-client and labels-at-server, compared with some baselines. The experimental results are positive. However, there are some minor issues in the experiment section. The authors mentioned that the proposed model can work under the scenario that some labels are not correct (noisy labels). However, this point was not verified in the experiment section at different error labeling rates. It is shown that FedProx-UDA/FixMatch degrade when the labeled data increases from 5 to 10 in Figure 6(d). What causes this degradation and how the proposed method avoid it?

---

> ### Author Response · Authors · 2020-11-15
> **Response to Reviewer #3**
>
> We sincerely appreciate your time and effort in reviewing our paper. We respond to your comments below:
>
> (1) There are some minor issues in the experiment section. The authors mentioned that the proposed model can work under the scenario that some labels are not correct (noisy labels). However, this point was not verified in the experiment section at different error labeling rates.
>
> - We double-checked our paper, but we were not able to find a scenario of noisy labels that you pointed out. Could you let us know what line numbers you refer to? We only introduce two realistic scenarios, which are labels-at-client and labels-at-server scenarios.
>
>     1. **Labels-at-Client** scenario: clients learn on both labeled and unlabeled data, while the global server only aggregates learned knowledge from the clients.
>     2. **Labels-at-Server** scenario: clients learn on unlabeled data, while the global server learns on labeled data.
>
> - The proposed scenario with noisy labels seems highly practical, but seems like an orthogonal problem to the federated semi-supervised learning problem we tackle. Yet, we expect our method to be more robust than naive baselines since we have inter-client consistency which pseudo-labels by maximizing agreement. We will evaluate on this scenario if time allows.
>
> ---
>
> (2) It is shown that FedProx-UDA/FixMatch degrade when the labeled data increases from 5 to 10 in Figure 6(d). What causes this degradation and how the proposed method avoid it?
>
> - This is an insightful comment. The base models suffer from catastrophic forgetting of the knowledge learned on labeled data, as they learn on unlabeled data with pseudo-labels. We conjecture that when the labeled data is small, unlabeled becomes relatively important as we can obtain limited information from labeled data, while it is less effective and even harmful when the amount of labeled data is sufficient. Our FedMatch on the other hand is not affected by the ratio of labeled and unlabeled samples, since it performs **disjoint learning**of parameters for labeled and unlabeled data, with **parameter decomposition**.
>
> - We provide further experimental evidence of our conjecture by additionally experimenting on data with different number of labels per class, using our method without the decomposition technique. Below are the new results:
>
> |             Number of Labels per Class              |     1    |     5    |    10    |    20    |
> |:---------------------------:|:--------:|:--------:|:--------:|:--------:|
> | Methods                     | Acc. (%) | Acc. (%) | Acc. (%) | Acc. (%) |
> | FedProx-UDA                 |   31.95  |   47.45  |   41.4   |   47.15  |
> | FedProx-FixMatch            |   30.01  |   47.2   |   34.25  |   44.5   |
> | FedMatch w/o Decomposition  |   **37.7**   |   47.51  |   51.15  |   62.7   |
> | FedMatch                    |   37.65  |   **54.5**   |   **60.65**  |   **66.1**   |
>
> - As shown in the above table, FedMatch without decomposition underperforms FedMatch with decomposition when the number of labels per class increases from 5 to 10 (around 3.x%p) than from 1 to 5 (around 9.x%p) or from 10 to 20 (10.x%p), similarly to baselines.
>
> - Contrarily, our method with the decomposition technique shows consistent performance improvements with any number of labeled samples. These results can be clear evidence that our decomposition technique is effective in preserving reliable knowledge from labeled data.
>
> - We believe that this result will further strengthen our paper as it is another evidence that shows the effectiveness of disjoint learning, and have included it in the Section C.2 of the Appendix, in the revision. Thank you for the insightful comment.

---

> ### Author Response · Authors · 2020-11-24
> **Additional experimental results under a noisy label scenario.**
>
> (1) There are some minor issues in the experiment section. The authors mentioned that the proposed model can work under the scenario that some labels are not correct (noisy labels). However, this point was not verified in the experiment section at different error labeling rates.
>
> - We thank you for the scenario that you suggested, where **some labels are not correct (noisy labels)**. It seems highly practical as one of the realistic federated semi-supervised learning scenarios, since it posits that some users might not correctly annotate all the locally-generated data, and thus there could exist some error rate on the local labeled data.
>
> - As suggested, we additionally conducted experiments for the **noisy-labels scenario**, as an extension of labels-at-client scenario, while increasing the ratio of noisy labels from 0.1 to 0.3. We validated on the CIFAR-10 dataset and generated Batch-IID tasks for 100 clients (0.05 communication faction used). We used 10 labeled examples per class for each client, while assigning random labels based on the noisy label ratio (excluding the original labels) for each class. We compare our model with FedProx-UDA/FixMatch during 100 training rounds. The result is as follows:
>
> | Ratio of Noisy Labels |   10 %   |   20 %   |   30 %   |
> |:---------------------:|:--------:|:--------:|:--------:|
> | Methods               | Acc. (%) | Acc. (%) | Acc. (%) |
> | FedProx-UDA           |   41.01  |   42.11  |   38.52  |
> | FedProx-FixMatch      |   43.39  |   40.09  |   39.25  |
> | **FedMatch (Ours)**       |   **45.95**  |   **45.31**  |   **42.51**  |
>
> - As shown in the above table, our method consistently shows 2.x%p - 4.x%p higher performances over base models across all noisy label ratios. Our method is more robust than naive baselines since our inter-client consistency performs pseudo-labeling by maximizing agreement across multiple models, learned at different clients. Our method thus can effectively alleviate performance degeneration caused by noisy labels, over baseline approaches.

---

### Official Review · AnonReviewer4 · 2020-11-03
**Interesting new federated learning scenario**

**Rating:** 6
**Confidence:** 3

**Review:**

The authors propose a new semi-supervised federated learning algorithm (FedMatch). Two scenarios are studies: 1) label-at-client (labeled and unlabeled data are at client). 2) label-at-server (labels are at server and unlabeled data are on client). The authors propose a disjoint learning which decomposes the model \theta into supervised \sigma and unsupervised $\psi$ such that $\theta = \sigma + \psi$ . the model is trained, in an alternating manner, to minimize the two loss functions for the supervised eq (4) and unsupervised model components eq (5) with inter-client consistency. The experimental results show an improvement w.r.t. to SOTA in terms of performance.

The paper is sometime difficult and its clarity could be improved. I would consider the novelty of the method, from machine learning prospective, to be somewhat borderline as it uses known elements and adapt them to the introduced learning scenarios but overall the proposed solution is interesting as it seems to work well in practice.

some detail comments:

- $\pi$ in one of the terms in the norm needs to be removed in inter-class consistency loss ||p θ (y|π(u)) − p θ (y|π(u))||
- The disjoint learning scenario was not well introduced in paragraph "Parameter Decomposition for Disjoint Learning"
- a missing section reference in (see section ??).
- Related to the reduction of communication costs: the bit-encoding of the model difference between the server and client is not clarified, i.e., for example if the difference is coded using 64 bits and also the same encoding for the models there is no gain in communication.
- In Table 1, the bold should be indicative of the best method per column

---

> ### Author Response · Authors · 2020-11-15
> **Response to Reviewer #4**
>
> We sincerely thank you for your time and effort in reviewing our paper. We respond to individual comments below:
>
> (1) The paper is sometime difficult and its clarity could be improved
>
> - To further improve the clarity of our paper, we have completely reorganized our paper by sequentially introducing the two different scenarios ("Labels-at-Client" and "Labels-at-Server") rather than elaborating on them simultaneously.
>
> - We have also corrected the typos and minor errors you pointed out in the revision (missing reference, typos, some unclear descriptions). We will be grateful if you check our revision for the updates.
>
> ---
>
> (2) I would consider the novelty of the method, from machine learning prospective, to be somewhat borderline as it uses known elements and adapt them to the introduced learning scenarios but overall the proposed solution is interesting as it seems to work well in practice."
>
> * We appreciate that you find our work interesting. We would like to recap our contributions, as we believe that our work is highly novel in multiple aspects:
>
> - **1) Novel problem:** We believe that our problem setting is highly novel, and is not a straightforward extension of semi-supervised learning in a federated learning setting, since "Labels-at-Server" scenario, where the labels are only available at the server, while clients only receive unlabeled data, is a unique but a realistic setting for federated learning.
>
> - **2) Inter-client consistency:** The inter-client consistency term is novel since existing approaches enforce consistency across predictions on the perturbations of the same sample (generated using data augmentation), while we enforce consistency across predictions from **multiple models**. Further, we do not straightforwardly enforce consistency across all models learned at all clients, since they may learn on heterogeneous data. Rather, we consider the **task relevance** to find the models that are helpful for a given task, by **embedding the models based on their functional similarities**and retrieving the nearest neighbors in the function space (Please see the last paragraph of Inter-Client Consistency part in the methodology).
>
> - **3) Disjoint learning:** To our knowledge, our is the first work that **decomposes the parameters for labeled and unlabeled data**for semi-supervised learning, and this has effectively **prevented**the unreliable knowledge from pseudo-labels from **negatively affecting**the reliable knowledge captured with labeled data. We believe that this is an important contribution, since catastrophic forgetting of the knowledge of labeled data is more severe in federated learning scenarios, where the models train on sequential data rather than on batch data. The proposed disjoint learning also provides a natural solution to "Labels-at-Server" scenario, where we need to combine knowledge of the labeled data learned at the server, with the knowledge of the unlabeled data learned at the clients.
>
> ---
>
> (3) Detailed Comments
> * We thank you so much for the detailed and helpful comments. We have reflected the corrections (missing reference, typos, some unclear descriptions) in the **revision** as follows:
> - We removed π in one of the terms as you mentioned
> - We added detailed algorithms and corresponding illustrative running examples to complement the explanation of "Parameter Decomposition for Disjoint Learning"
> - We corrected a missing section reference that you mentioned.
> - Regarding the reduction of communication costs, it is important that we reduce the amount of information that needs to be sent between both server and client. The actual bit-level compression techniques are implementation problems, which are beyond our research topic. However, we added this assumption into our new revision.
> - We bold our methods, since the SL (Supervised Learning) models are not Federated Semi-Supervised Learning (FSSL) algorithms. We include them as our upper bound since they learn with full labels. We rather emphasize “SL models learn with full labels“ in captions of both Table 1 and 2.

---

### Author Response · Authors · 2020-11-18
**Summary of the  Initial Revision**

We sincerely thank all reviewers for your constructive and helpful comments. During the rebuttal period, we have revised our paper to faithfully reflect **all the comments**from the reviewers, conducting **multiple sets of experiments**. We have made the following updates to the revision.

1. We have completely reorganized our paper by **sequentially introducing the two different scenarios** ("Labels-at-Client" and "Labels-at-Server") to improve the clarity and readability of our paper (please see Section 4 for labels-at-client and Section 5 for labels-at-server scenario).
2. We have conducted experiments on the **COVID-19 Radiography Database, which is a real-world dataset**that consists of X-ray images from COVID and non-COVID patients (please see Section C.1 in the Appendix).
3. We have included additional experiments to show that our method scales to **a larger number of classes per class** (please see Section C.2 in the Appendix).
4. We have moved the **pseudo-code algorithms**for both scenarios back into the **main paper** (Algorithm 1 in page 5, Algorithm 2 in page 6) from the Appendix.
5. We have corrected the typos, errors, missing and incorrect references pointed out by the reviewers (colored with blue)

We believe that our revised version of the paper has been largely improved in terms of the clarity and readability, and have become stronger with the additional experimental results, thanks to your helpful suggestions. We want to emphasize again that exploring the unique challenges of federated semi-supervised learning in two realistic scenarios (labels-at-client and labels-at-server), as well as the proposal of novel technical methods to tackle them (**inter-client consistency regularization** and **disjoint learning with decomposed parameters**) are both highly novel contributions with large practical impact.

---

### Comment · ~Jieming_Bian1 · 2021-08-29
**Questions about the experiment results**

Dear authors,

I got the following two questions about your experiment results.

1. In this paper, for the Cifar-10 Batch-IID and Batch Non-IID experiments on label-at-server scenario,  you described about total 5,000 labeled data located at server while 49,000 unlabeled data stored in 100 clients, and the result of FedMatch is 44.95% for Batch-IID and 44.17% for Batch Non-IID. However, in the code you provided in you Github, the total labeled data utilized in the experiment is 1000. I have tested the FedMatch with 1000 labeled data in server, and get the similar result like you provided in this paper which is around 45%. But when I change the number of labeled data to 5000 which is the number you state in the paper, the result becomes around 55%. So does it mean you actually get the result with 1000 labeled data instead of 5000 labeled data?

2. I also find that you didn't provide an important baseline result which is the supervised learning trained with the only labeled data located in server. I have tried to train with the only 5000 labeled Cifar-10 data (500 for each class) stored in server with backbone model is also Resnet 9. The accuracy can be around 70% which is much higher than 55% which trained by both 5000 labeled server data and 49000 unlabeled client data with utilizing your FedMatch method. Is there something wrong with my experiment? Can you please provide this baseline result?

Thanks!

---

> ### Author Response · Authors · 2021-08-29
> **Answers**
>
> Dear Jieming,
>
> Thank you for your questions. Let me answer them.
>
> ---
>
> **“In the code you provided in you Github, the total labeled data utilized in the experiment is 1000. does it mean you actually get the result with 1000 labeled data instead of 5000 labeled data?”**
> - **1,000 labeled data is correct**. In Section A.3 and Table 4 (training details), we specified that **LPC (the number of labeled examples per class) is 100**, yielding 1,000 labeled data at the server (100 labeled examples * 10 classes). Hence, as described, we used 1,000 labeled data at the server under the Labels-at-Server scenario, which our code and experimental result are also correct.
> - In Section 6.2, although we described the task configuration is based on the Labels-at-Client scenario, we see that it might be confusing when it comes to the Labels-at-Server scenario. We’ll move some details for Labels-at-Server configurations from Appendix to Section 6.2 of main document. Thank you for pointing it out.
> - Thank you for your additional experiments. Performance increase seems natural with 5 times more labeled data.
>
> ---
> **“I also find that you didn't provide an important baseline result which is the supervised learning trained with the only labeled data located in server. I have tried to train with the only 5000 labeled Cifar-10 data (500 for each class) stored in server with backbone model is also Resnet 9. The accuracy can be around 70% which is much higher than 55% which trained by both 5000 labeled server data and 49000 unlabeled client data with utilizing your FedMatch method. Is there something wrong with my experiment? Can you please provide this baseline result?**
> - We provided three supervised-learning baselines, including **Local-SL**, **FedAvg-SL**, and **FedProx-SL**, that are trained on **not only **$S$**, but also **$U$** with corresponding labels**, which are more powerful baselines compared to one trained on only **$S$**.
> - We double-checked the performance of SLs for the Labels-at-Server scenario and reproduced the same accuracy ranges (around 50-51) reported in the our paper with the code uploaded on GitHub.
> - As you asked, we also conducted SL on only **$S$** at the server under the Labels-at-Server scenario. We observed accuracies in a range of 43.85 - 44.0, which are much lower compared to the original baselines trained on **$S$** and **$U$** (around 50-51).
>
> ---
> Thank you again for your interest in our work, and I hope these answers help you.
>
> Best regard,
> The authors.

---

> > ### Comment · ~Jieming_Bian1 · 2021-08-29
> > **The experiment result of SL on only S is a lower bond.**
> >
> > Dear author,
> >
> > Thanks so much for your reply. I am still confused with the second point. I do agree with the three supervised-learning baselines(Local-SL, FedAvg-SL and FedProx-SL) you provided. But you method FedMatch does not outperform any of the three baselines. These three baselines could be the upper bond since they utilize the unlabeled data as labeled. But, the SL on only S at the server is non-trivial. It is the lower bond. As you said, the accuracies of this SL is in a range of 43.85%-44.0%. However, your method FedMatch only get the accuracies as 44.17% (Non-IID) and 44.95% (IID) with both the labeled data at server and unlabeled data at clients. It seems that the improvement is so small (less than 1%) as you utilizing 54,500 more unlabeled data.
> >
> > Best,
> > Jieming

---

> > > ### Author Response · Authors · 2021-08-30
> > > **Response**
> > >
> > >
> > > Dear Jieming,
> > >
> > > Thank you for your questions. Here are our responses.
> > >
> > > ---
> > > **“I am still confused with the second point. I do agree with the three supervised-learning baselines(Local-SL, FedAvg-SL and FedProx-SL) you provided. But you method FedMatch does not outperform any of the three baselines. These three baselines could be the upper bond since they utilize the unlabeled data as labeled.“**
> > >
> > > - We never claimed that FedMatch outperforms the supervised learning baselines which assume that GT labels exist for the unlabeled data. Rather, we repeatedly mentioned and clearly emphasized in our paper that the three baselines are **our upper bounds** -- “Note that the SL (Supervised Learning) models learn on both $S$ and $U$ with full labels, and are utilized as the upper bounds for each experiment.” -- in the captions of Table 1 and Figure 5 (Page 7) and Table 2 (Page 8).
> > >
> > > ---
> > >
> > > **”But, the SL on only S at the server is non-trivial. It is the lower bond. As you said, the accuracies of this SL is in a range of 43.85%-44.0%. However, your method FedMatch only get the accuracies as 44.17% (Non-IID) and 44.95% (IID) with both the labeled data at server and unlabeled data at clients. It seems that the improvement is so small (less than 1%) as you utilizing 54,500 more unlabeled data.”**
> > >
> > > - We do not agree that the model which only utilizes the labels at the server is a lower bound. As you can see from the experimental results in the paper, the naive combination of federated learning with semi-supervised learning actually underperforms the baseline you mentioned, as they suffer from catastrophic forgetting, since the unreliable knowledge from unlabeled data negatively affects the SSL model’s performance. Thus, a semi-supervised learning method which utilizes unlabeled local data could underperform the model which only utilizes the labeled data at the server, and the baseline is not a lower bound.
> > >
> > > - You mentioned that the improvements are ‘small’, but this is a subjective argument. Whether the improvements are small or large should be determined based on the comparison to relevant baselines, and both the local semi-supervised learning and a naive combination of federated learning with SSL largely underperform both the SL at the server baseline, our FedMatch. For the Labels-at-Server scenario, fixing such failures of the baseline SSL method was the main goal, and thus we do not believe that the performance gains are small.
> > >
> > > - Finally, we can obtain higher performance gains with SSL by decreasing $\lambda_{L_1}$, which controls the sparsity of $\psi$. Note that we have set the sparsity to be very high, in order to reduce the S2C (server-to-client) and C2S (client-to-server) communication costs. When setting the sparsity to 0, our FedMatch achieves the accuracy of 45.76 on the Non-IID dataset, and 45.41 on the IID dataset. Note that the communication cost will be still smaller than the baselines, since it converges faster. We report 100-round accuracy on Labels-at-Server scenario in the below table. The experimental setup is the same as described in the paper except for 0 for $\lambda_{L_1}$, 0.05-0.1 for $\lambda_{L_2}$, 0.01 for $\lambda_{L_{iccs}}$, 0.1-1.0 for $\lambda_{L_{cons}}$, and soft augmentation for local consistency loss.
> > >
> > >
> > > | Batch-NonIID    |               |
> > > |-----------------|---------------|
> > > | Method          | Accuracy      |
> > > | FedAvg-SL (S+U) | 51.34 ± 0.42  |
> > > | FedMatch        | 45.76± 0.32	   |
> > > | SL (S)          | 37.28 ± 2.34  |
> > >
> > > | Batch-IID    |              |
> > > |-----------------|--------------|
> > > | Method          | Accuracy     |
> > > | FedAvg-SL (S+U) | 49.95 ± 0.17 |
> > > | FedMatch        | 45.41 ± 0.29 |
> > > | SL (S)          | 37.28 ± 2.34 |
> > >
> > > Best,
> > > Authors

---

### Decision · Program_Chairs · 2021-01-07
**Final Decision**

**Decision:**

Accept (Poster)

**Comment:**

This work proposes the Federated Matching algorithm as a novel method to tackle the problems in federated learning. The paper is well-written and original, and it contributes to the state-of-the-art.